# Neurostimulation in People with Oropharyngeal Dysphagia: A Systematic Review and Meta-Analysis of Randomised Controlled Trials—Part II: Brain Neurostimulation

**DOI:** 10.3390/jcm11040993

**Published:** 2022-02-14

**Authors:** Renée Speyer, Anna-Liisa Sutt, Liza Bergström, Shaheen Hamdy, Timothy Pommée, Mathieu Balaguer, Anett Kaale, Reinie Cordier

**Affiliations:** 1Department Special Needs Education, University of Oslo, 0318 Oslo, Norway; anett.kaale@isp.uio.no; 2Curtin School of Allied Health, Faculty of Health Sciences, Curtin University, Perth, WA 6102, Australia; reinie.cordier@northumbria.ac.uk; 3Department of Otorhinolaryngology and Head and Neck Surgery, Leiden University Medical Centre, 1233 ZA Leiden, The Netherlands; 4Critical Care Research Group, The Prince Charles Hospital, Brisbane, QLD 4032, Australia; annaliisasp@gmail.com; 5School of Medicine, University of Queensland, Brisbane, QLD 4072, Australia; 6Remeo Stockholm, 128 64 Stockholm, Sweden; liza.bergstrom@regionstockholm.se; 7Speech Therapy Clinic, Danderyd University Hospital, 182 88 Stockholm, Sweden; 8Faculty of Biology, GI Sciences, School of Medical Sciences, Medicine and Health, University of Manchester, Manchester M13 9PL, UK; shaheen.hamdy@manchester.ac.uk; 9IRIT, CNRS, Université Paul Sabatier, 31400 Toulouse, France; timothy.pommee@irit.fr (T.P.); mathieu.balaguer@irit.fr (M.B.); 10Norwegian Centre of Expertise for Neurodevelopmental Disorders and Hypersomnias, Oslo University Hospital, 0424 Oslo, Norway; 11Department of Social Work, Education and Community Wellbeing, Faculty of Health & Life Sciences, Northumbria University, Newcastle upon the Tyne NE7 7XA, UK

**Keywords:** deglutition, swallowing disorders, RCT, intervention, repetitive transcranial magnetic stimulation, transcranial direct current stimulation, rTMS, tDCS

## Abstract

*Objective.* To assess the effects of brain neurostimulation (i.e., repetitive transcranial magnetic stimulation [rTMS] and transcranial direct current stimulation [tDCS]) in people with oropharyngeal dysphagia (OD). *Methods.* Systematic literature searches were conducted in four electronic databases (CINAHL, Embase, PsycINFO, and PubMed) to retrieve randomised controlled trials (RCTs) only. Using the Revised Cochrane risk-of-bias tool for randomised trials (RoB 2), the methodological quality of included studies was evaluated, after which meta-analysis was conducted using a random-effects model. *Results.* In total, 24 studies reporting on brain neurostimulation were included: 11 studies on rTMS, 9 studies on tDCS, and 4 studies on combined neurostimulation interventions. Overall, within-group meta-analysis and between-group analysis for rTMS identified significant large and small effects in favour of stimulation, respectively. For tDCS, overall within-group analysis and between-group analysis identified significant large and moderate effects in favour of stimulation, respectively. *Conclusion.* Both rTMS and tDCS show promising effects in people with oropharyngeal dysphagia. However, comparisons between studies were challenging due to high heterogeneity in stimulation protocols and experimental parameters, potential moderators, and inconsistent methodological reporting. Generalisations of meta-analyses need to be interpreted with care. Future research should include large RCTs using standard protocols and reporting guidelines as achieved by international consensus.

## 1. Introduction

Oropharyngeal dysphagia (OD) or swallowing problems is highly prevalent among stroke patients, people with progressive neurological diseases, patients with head and neck cancer, and in frail older persons [1,2]. Prevalence estimates of OD may vary depending on underlying medical diagnoses, but have been reported as high as 80% in stroke and Parkinson’s disease [3], and 70% in oncological populations [4]. OD is associated with dehydration, malnutrition, aspiration pneumonia, and increased mortality [5,6,7], but also leads to decreased health-related quality of life [8].

Treatment and management of OD may vary widely. However, apart from traditional compensatory and rehabilitative strategies including diet modifications, postural adjustments, oromotor training and swallow manoeuvres [9], recent studies report on the possible beneficial effects of non-invasive brain stimulation. Brain neurostimulation aims to modulate cortical excitability and include techniques such as repetitive transcranial magnetic stimulation (rTMS) and transcranial direct current stimulation (tDCS). rTMS uses electromagnetic induction resulting in depolarisation of postsynaptic connections, whereas tDCS uses direct electrical current shifting the polarity of nerve cells [10]. Neurostimulation protocols may vary greatly per study, including different neurostimulation sites, frequencies, stimulation duration and number of different outcome measures are used to objectify treatment effects, and individual responses to stimulation are highly variable [10,11,12].

Aspiring to improved treatment efficacy in OD management, non-invasive brain stimulation has achieved growing interest over the past decade. Several reviews have been published on rTMS and tDCS [10,12,13,14,15,16,17,18], each publication having different inclusion and exclusion criteria and methodology. All previous reviews targeted brain neurostimulation interventions in post-stroke populations except for one review that included patients with acquired brain injury [16]; to date, all reviews on brain stimulation set criteria based on medical diagnoses. Moreover, not all reviews performed meta-analysis [14] and as several neurostimulation trials have only been published recently, earlier reviews will have identified fewer studies.

This is the second paper (Part II) of two companion papers on treatment effects of neurostimulation in people with OD. The first systematic review (Part I) reported on the effects of pharyngeal electrical stimulation (PES) and neuromuscular electrical stimulation (NMES).

The aim of this systematic review (Part II) is to determine the effects of brain neurostimulation (i.e., rTMS and tDCS) in people with OD without excluding populations based on medical diagnoses. Only randomised controlled trials (RCTs) will be included being the highest level of evidence. Meta-analyses will be conducted to summarise results and report on possible moderators of treatment effects.

## 2. Methods

The methodology and reporting of this systematic review followed the Preferred Reporting Items for Systematic Reviews and Meta-Analyses (PRISMA) 2020 statement and checklist (Appendix A) [19,20]. Adhering to the PRISMA statement and checklist ensures essential and transparent reporting of systematic reviews. The protocol for this review was registered with PROSPERO, the international prospective register of systematic reviews (registration number: CRD42020179842).

### 2.1. Information Sources and Search Strategies

An electronic database search for extant literature was conducted on 6 March 2021, using the following four databases: CINAHL, Embase, PsycINFO, and PubMed. Publications dates included in the search were 1937–2021, 1902–2021, 1887–2021, and 1809–2021, respectively. Generally, search strategies consisted of combinations of terms related to ‘dysphagia’ and ‘randomised controlled trial’. Both subject headings (e.g., MeSH and Thesaurus terms) and free text terms were used to search databases. The full list of electronic search strategies for each database can be found in Table 1. To identify literature not found utilising these strategies, the reference lists of eligible articles were checked.

### 2.2. Inclusion and Exclusion Criteria

To be eligible for inclusion in this systematic review, studies had to meet the following criteria: (1) participants had a diagnosis of oropharyngeal dysphagia; (2) the study included non-invasive neurostimulation interventions aimed at reducing swallowing or feeding problems; (3) the study included a control group or comparison intervention group; (4) participants were randomly assigned to one of the study arms or groups; and (5) the study was published in English language.

Interventions such as non-electrical peripheral stimulation (e.g., air-puff or gustatory stimulation), pharmacological interventions and acupuncture, were considered out of scope of this review, thus were excluded. Invasive techniques and/or those that did not specifically target OD (e.g., deep-brain stimulation studies after neurosurgical implementation of a neurostimulator) were also excluded. Conference abstracts, doctoral theses, editorials, and reviews were excluded.

### 2.3. Systematic Review

*Methodological Quality and Risk of Bias.* The Revised Cochrane risk-of-bias tool for randomised trials (RoB 2) [21] was used to assess the methodological quality of the included studies. The RoB 2 tool identifies domains to consider when assessing where bias may have been introduced into a randomised trial: (1) bias arising from the randomisation process; (2) bias due to deviations from intended interventions; (3) bias due to missing outcome data; (4) bias in measurement of the outcome; and (5) bias in selection of the reported result. For each domain, a series of signalling questions are answered to give a judgement (i.e., “low risk of bias”, “some concerns”, or “high risk of bias”), which can then be assessed in aggregate to determine a study’s overall risk of bias [21].

*Data Collection Process.* Data were extracted from the included studies using a data extraction form created for this purpose. This form allowed for extraction of data under several categories, relevant to meta-analyses, including participant diagnosis, inclusion and exclusion criteria, sample size, age, gender, intervention goal, intervention agent/delivery/dosage, outcome measures, and treatment outcomes.

*Data, Items and Synthesis of Results.* Titles and abstracts of included studies were reviewed for eligibility by two independent reviewers. Next, the same two reviewers assessed the selected original articles at a full-text level to determine their eligibility. To ensure rating accuracy, a random selection of one hundred records were scored and discussed over two consecutive group sessions prior to rating the remaining records. Any disagreement between the first two reviewers was resolved by consulting a third reviewer. Assessment of methodology study quality followed an equivalent process. None of the reviewers had formal or informal affiliations with any of the authors of the included studies.

Extracted data were extrapolated and synthesised within the following categories to allow for comparison: participant characteristics, inclusion criteria, intervention conditions, outcome measures and intervention outcomes. Effect sizes and significance of findings were used to assess treatment outcomes.

### 2.4. Meta-Analysis

Using the extracted data, effect sizes were compared for the following: (1) pre-post outcome measures of OD and (2) mean difference in outcome measures from pre- to post-intervention scores between neurostimulation and comparison controls. Control groups either received no treatment, sham stimulation and/or traditional dysphagia therapy (DT; e.g., compensatory and rehabilitative strategies including diet modifications, postural adjustments, oromotor training and swallow manoeuvres). Only studies using instrumental assessment (e.g., videofluoroscopic swallow study [VFSS] or fiberoptic endoscopic evaluation of swallowing [FEES]) to confirm OD were included.

When selecting what data points to extract, data collected using outcome measures based on visuoperceptual evaluation of instrumental assessment were preferred over clinical non-instrumental assessments. Oral intake measures were only included if no other clinical data were available, whereas screening tools and patient self-report measures were excluded entirely. When selecting outcome measures for meta-analyses, reducing heterogeneity between studies was given priority. Consequently, measures other than the authors’ primary outcomes may have been preferred if these measures contributed to greater homogeneity.

Comprehensive Meta-Analysis Version 3.3.070 [22] software was used to complete the meta-analysis, allowing comparison of sample size, effect size, group means and standard deviations of pre- and post-measurements. In the case that no parametric data were available, the reported non-parametric data (i.e., medians, interquartile ranges) were converted into parametric data for meta-analysis purposes. Studies with multiple intervention groups were analysed separately for each experimental-control comparison. If studies included the same participants, only one study was included in the meta-analysis. Where reported data were insufficient, attempts were made to contact authors of individual studies and request additional data.

Using Comprehensive Meta-Analysis, a random-effects model was used to calculate effect sizes. This was due to variations in participant characteristics, sampling, interventions, and measurement, which suggested a low likelihood that studies would have similar true effects. Heterogeneity was estimated using the *Q* statistic to determine the spread of effect sizes about the mean and *I*^2^ was used to estimate the ratio of true variance to total variance. *I*^2^-values of less than 50%, 50% to 74%, and higher than 75% denote low, moderate, and high heterogeneity, respectively [23]. Effects sizes were generated using the Hedges’ *g* formula for standardised mean difference with a confidence interval of 95%. Effects sizes were interpreted using Cohen’s *d* convention as follows: *d* ≤ 0.2 as no or negligible effect; 0.2 < *d* ≤ 5 as small effect; 0.5 < *d* ≤ 0.8 as moderate effect; and *d* > 0.8 as large effect [24].

Forest plots of effect sizes for OD outcome scores were generated for both types of neurostimulation (i.e., rTMS and tDCS): (1) pre-post neurostimulation and (2) neurostimulation interventions versus comparison groups. Subgroup analyses were conducted to compare effect sizes as a function of different moderators and neurostimulation types including: outcome measures, total treatment duration, total neurostimulation time, and stimulation characteristics (e.g., pulse range, stimulation current, and stimulation site). To take into consideration the possibility of spontaneous recovery during the intervention period, only between-subgroup meta-analyses were conducted using post-intervention data.

Utilising Comprehensive Data Analysis software, publication bias was evaluated as per the Begg and Muzumdar’s rank correlation test and the Fail-safe N test. Begg and Muzumdar’s rank correlation test provides information on the rank correlations between standardised effect size and the ranks of their variances [25]. In addition to a tau value, a two-tailed *p* value is also generated. Where the analysis results in a value of zero, it can be concluded that there is unlikely to be an association between the effect size and ranks of variance. Conversely, the closer to one the tau or *p* values, the more likely there is to be an association between the effect size and ranks of variance. Therefore, high standard error would be connected to higher effect sizes if publication bias was the result of asymmetry. If larger effects are represented by low values, tau would be over zero; conversely tau would be negative if larger effects are represented by high values.

The Fail-safe N test is a calculation of the quantity of studies with zero effect size that could be incorporated into the meta-analysis prior to the result losing statistical significance, that is, the quantity of excluded studies that would result in the effect being nullified [26]. Results should be treated with care where the fail-safe N is relatively small, however, when it is large, conclusions can be confidently drawn that the treatment effect, while potentially raised by the removal of some studies, is not nil.

## 3. Results

### 3.1. Study Selection

A total of 8059 studies were retrieved through the subject heading and free text searches (CINAHL: *n* = 239, Embase: *n* = 4550, PsycINFO: *n* = 231, and PubMed: *n* = 3039). Following removal of duplicates at a title and abstract level (*n* = 1113), a total of 6946 records remained. A total of 261 original articles were assessed at a full-text level, with articles grouped according to type of intervention. At this stage, no studies were excluded based on type of intervention (e.g., behavioural intervention, neurostimulation). Of these, 58 articles on neurostimulation were identified that satisfied the inclusion criteria. Four additional studies were found through reference checking of the included articles. This process resulted in a final number of 24 included studies. Figure 1 presents the flow diagram of the overall reviewing process according to PRISMA.

### 3.2. Description of Studies

Table 2 and Table 3 report detailed descriptions of all included studies. Table 2 includes data on study characteristics including methodological study quality, inclusion and exclusion criteria, and details on participant groups. Information is provided for all study groups (control and intervention groups), medical diagnosis, sample size, age and gender. Table 3 reports on intervention characteristics, including goals, intervention components, outcome measures, intervention outcomes, as well as main conclusions.

*Brain stimulation Interventions* (Table 2). Across the 24 included studies, eleven studies reported on rTMS and nine studies reported on tDCS. Four studies used another type of neurostimulation (i.e., NMES) in addition to rTMS, either within the same group or over different treatment groups.

*Participants* (Table 2). The 24 studies included a total of 728 participants (mean 30.3; SD 13.4). The sample sizes ranged from the smallest sample of 14 participants [27] to the largest sample of 64 participants [28]. By intervention type, samples were characterised as follows: *rTMS* total 280, mean 25.5, SD 7.6, range 15–40; *tDCS* total 283, mean 31.4, SD 14.6, range 14–59; and combined neurostimulation total 165, mean 41.3, SD 19.3, range 18–64. The mean age of participants across all studies was 64.6 years (SD 5.8), ranging from 51.8 years [29] to 74.9 years [27]. By intervention group, the mean age of participants was: *rTMS* 63.6 (4.8), *tDCS* 66.2 (SD 6.9), and combined neurostimulation 66.5 years (SD 4.4).

Across all studies 59.6% (SD 12.7) participants were male and two studies did not report gender distribution [29,30]. Percentage of males by intervention group was *rTMS* 61.9% (SD 12.8), *tDCS* 57.5% (SD 10.9), and other/combined 65.4% (SD 12.3). Most studies included stroke patients (*n* = 21), with other diagnoses by intervention group reported as: presbyphagia due to central nervous system disorder (*n* = 1) [31] in *tDCS*; Parkinson’s disorder (*n* = 1) [30] and brain injury (*n* = 1) [32] in *rTMS*. All 24 studies used VFSS to confirm participants’ diagnosis of OD. The studies were conducted across 12 countries, with the highest number of studies conducted in Korea (*n* = 6), Egypt (*n* = 4), China (*n* = 3), Italy (*n* = 2) and Japan (*n* = 2).

*Outcome Measures* (Table 3). Outcomes measures varied greatly across all studies included in the review, covering several domains within the area of OD. The Penetration Aspiration Score (PAS) was the most reported outcome measure (8 studies), followed by the Dysphagia Outcome and Severity Scale (DOSS; 7 studies), Functional Oral Intake Scale (FOIS; 3 studies) and Degree of Dysphagia (DD; 3 studies).

*rTMS Intervention (n = 11:*Table 2 and Table 3*).* All but one of the rTMs studies [33] compared rTMS stimulation with sham rTMS. One single study compared rTMS with rTMS combined with DT, and DT only [33]. Three more studies included three arms; two studies compared rTMS using different frequencies versus sham rTMS [32,34], and one study compared bilateral and unilateral rTMS versus sham rTMS [35].

*tDCS Intervention (n = 9:*Table 2 and Table 3*).* Eight studies compared tDCS with sham tDCS [27,29,31,36,37,38,39,40,41], and one study compared tDCS with theta-burst stimulation (TBS) [31]. All but one study (31) combined both study arms with DT. In one study both groups received simultaneous catheter balloon dilatation in addition to DT [40].

*Combined Neurostimulation Interventions (n = 4: see*Table 2 and Table 3*).* Three studies in the combined intervention group compared three different treatments. Of these, one compared rTMS, PES and paired associative stimulation (PAS) [42], a second compared DT, rTMS combined with DT, and NMES combined with DT [43], and a third compared rTMS, PES and capsaicin stimulation [44]. A fourth study combined NMES stimulation with sham rTMS or rTMS stimulating different hemispheres (ipsilesional, contralesional or bilateral) [45].

**Table 2 jcm-11-00993-t002:** Study characteristics of studies on rTMS and tDCS interventions for people with oropharyngeal dysphagia.

Study Author (Year) Country	Inclusion/Exclusion Criteria	Sample (*n*) Groups	Group Descriptives (Mean ± SD) Age, Gender, Medical Diagnoses	Procedure, Delivery and Dosage per Intervention Group ^a^
**repetitive Transcranial Magnetic Stimulation (rTMS)—*n* = 11**				
Cheng et al. (2017) [46] Hong Kong, China	OD not defined. Screened face-to-face or via telephone using inclusion criteria Inclusion: chronic post-stroke (>12 months); ≤80 years; able to follow simple instructions and sit upright for 30 min Exclusion: previous history of epilepsy, dysphagia, head injury or other neurological disease; neurosurgery; oral/maxillofacial surgery; presence of magnetic implants; medically unstable and on medications that lower neural threshold	*n* = 15 Treatment group (11), 73.3% rTMS Sham group (4), 26.7% Sham rTMS	Treatment group: Age = 65.1 ± 8.3 Male = 64% Sham group: Age = 63.3 ± 7.8 Male = 100% NS difference between groups in age or post-stroke duration.	Procedure: rTMS (Magstim Rapid) daily for 10 days over 2 weeks rTMS (5 Hz) to the tongue area of the motor cortex of affected hemisphere, identified by MRI, via Magstim coil Treatment group: Thirty 100-pulse trains of 5 Hz rTMS, with inter-train interval of 15 s Stimulation at 90% of patient’s resting motor threshold Sham: rTMS via a sham Magstim coil (identical appearance and noise, but no active stimulation) Identical stimulation schedules
Du et al. (2016) [34] China	OD as per clinical assessment. Inclusion: first monohemispheric ischaemic stroke <2 months ago; single infarction Exclusion: other concomitant neurological disease; fever; infection; use of sedatives; severe aphasia or cognitive impairment; inability to complete follow-up; contraindications for stimulation used in study	*n* = 40 Treatment group 1 (15), 37.5% High frequency rTMS Treatment group 2 (13), 32.5% Low frequency rTMS Sham group (12), 30.0% sham rTMS	Treatment group 1: Age 58.2 ± 2.8 87% male Location of lesion: cortical (1), subcortical (10), massive (4) Treatment group 2: Age 57.9 ± 2.5 54% male Location of lesion: cortical (0), subcortical (9), massive (4) Sham group: Age 58.8 ± 3.4 50% male Location of lesion: cortical (2), subcortical (5), massive (5) NS differences between groups.	Procedure: rTMS (MagPro ×100 stimulator) targeting the mylohyoid cortical area of hemisphere (‘hot spot’), identified by EMG. Coil angle approximately = 45 degrees. Daily for 5 consecutive days Treatment group 1, high frequency stimulation: 3 Hz rTMS for 10 s Inter-train interval of 10 s, and 40 trains with a total of 1200 pulses at 90% rTMS on the *affected hemisphere* Treatment group 2, low frequency stimulation: 1 Hz rTMS for 30 s Inter-train interval of 2 s, and 40 trains with a total of 1200 pulses at 100% rMT on the *unaffected hemisphere* Sham: Similar conditions to Treatment group 2 to imitate noise of the stimulation with coil rotated 90 degrees away from the scalp
Khedr et al. (2009) [47] Egypt	OD as per swallowing questionnaire confirmed by bedside examination. Inclusion: single thromboembolic non-haemorrhagic infarction of the middle cerebral artery with acute hemiplegia and dysphagia Exclusion: unstable cardiac arrhythmia, fever, infection, hyperglacaemia, prior administration of sedatives, inability to give informed consent due to severe aphasia, anosognosia or cognitive deficits	*n* = 26 Treatment group (14), 53.8% rTMS Sham group (12), 46.2% Sham rTMS	Treatment group: Age 58.9 ± 11.7 Sham group: Age 56.2 ± 13.4 No group specific descriptors given. Overall, 38.5% male. 14 with right-sided hemiplegia and 12 patients with left-sided hemiplegia. NS difference between groups.	Procedure: rTMS or sham 5 consecutive days, 10 min at a time A total of 300 3 Hz rTMS pulses at an intensity of 120% resting motor threshold, delivered by Dantec Maglite (TM Copenhagen, Denmark). Figure-of-eight coil placed over oesophageal cortical area of the affected hemisphere, identified by EMG. Sham group: Similar parameters producing the same noise, but with the coil rotated away from scalp
Khedr and Abo-Elfetoh (2010) [48] Egypt	OD as per swallowing questionnaire and bedside swallow screening Inclusion: conscious patient within 1–3 months of first ever ischaemic stroke (LMI or other brainstem infarction with pontomedullary dysfunction); degree of dysphagia from grade III to IV Exclusion: head injury or other neurological disease than stroke; unstable cardiac arrhythmia; fever; infection; hyperglycaemia; epilepsy or prior administration of tranquilisers; presence of intracranial metallic devices or pacemakers; inability to give informed consent	*n* = 22 Treatment group (11), 50% rTMS Sham group (11), 50% Sham rTMS	Group statistics given based on infarction type divided into treatment versus sham. Lateral medullary infarction group: Treatment group (6): Age 56.7 ± 16 100% male Sham (5): Age: 58 ± 17.5 100% male Other brainstem infarction group: Treatment group (5): Age: 55.4 ± 9.7 40% male Sham (6): Age: 60.5 ± 11 50% male NS difference between groups.	Procedure: rTMS or sham 5 consecutive days for 10 min Treatment group: 10 trains of 10 s 3 Hz stimulation, repeated every minute, delivered by Mag-Lite r25 stimulator (Dantec Medical, Denmark). Intensity set at 130% of resting motor threshold Figure-of-eight coil placed over oesophageal cortical area of both hemispheres, judged to be about 3 cm anterior and 6 cm lateral to the vertex (neurophysiology explorations not performed on participants due to severity of vertigo and dysphagia). Sham group: Similar parameters producing the same noise, but with coil rotated away from scalp
Khedr et al. (2019) [30] Egypt	OD as per Swallowing Disturbance Questionnaire (SDQ) Inclusion: 50–75 years old patients with Parkinson’s Disease Exclusion: history of repeated head injury, cerebrovascular accident, encephalitis, oculogyric crisis, supranuclear gaze palsy, exposure to antipsychotics or MPTP (1-methyl-4-phenyl-1,2,3,6-tetrahydropyridine), severe dementia or depression, severe dysautonomia, cerebellar signs, Babiniski sign, strictly unilateral features after 3 years, hydrocephalus, intracranial lesion, contraindications to repetitive transcranial magnetic stimulation (rTMS), inability to give informed consent	*n* = 30 Treatment group (19), 63.3% rTMS Sham group (11), 36.7% Sham rTMS	Treatment group: Age 60.7 ± 8.8 duration of illness 5.7 +/− 3.9 Hoehn and Yahr 3.1 +/− 1.1 Sham group: Age 57.4 ± 10.0 duration of illness 6.5 +/− 3.7 Hoehn and Yahr 3.5 +/− 1.0 Gender distribution not given. NS difference between groups.	Procedure: rTMS or sham (Magstim 200) 10 days (5 days per week) followed by 5 booster sessions every month for 3 months 10 trains of 20 Hz stimulation, each lasting for 10 s with intertrain interval of 25 s. Intensity set at 90% of the RMT. Stimulation to cortical area: first dorsal interosseous (hand area) for each hemisphere. Location identified from where rTMS elicited MEPs of the highest amplitude. Both hemispheres stimulated, one at a time during each session. Sham group: Similar parameters producing the same noise, but with the coil rotated away from scalp
Kim et al. (2011) [32] Korea	OD as per VFSS Inclusions: dysphagia post-brain injury <3 months ago; unilateral hemisphere involvement Exclusion: prior neurological disease; unstable medical condition; severe cognitive impairment; severe aphasia; history of seizures	*n* = 30 Treatment group 1 (10), 33.3% High frequency rTMS Treatment group 2 (10), 33.3% Low frequency rTMS Sham group (10), 33.3% Sham rTMS	Treatment group 1: Age: 69.8 ± 8.0 50% male Stroke (9), TBI (1) Treatment group 2: Age: 66.4 ± 12.3 66.6% maleStroke (10), TBI (0) Sham group: Age: 68.2 ± 12.6 66.6% male Stroke (9), TBI (1) NS difference between groups.	Procedure: rTMS or sham (Magstim 200) using a figure-eight coil cooled with air Once a day for 20 min on 10 days (5 times a week for 2 weeks) All groups received DT, which included oral and facial sensory training, oral and pharyngeal muscle training, compensatory techniques, and NMES^b^ on pharyngeal muscles during rTMS. Stimulation sites identified by evaluation of MEPs of the bilateral mylohyoid muscles. Treatment group 1: High intensity rTMS Ipsilateral hemisphere hotspot at 100% of each MEP threshold At 5 Hz, for 10 s, and repeated every minute for 20 min (total 1000 pulses) Treatment group 2: Low intensity rTMS Contralesional hemisphere hotspot at 100% MT At 1 Hz for 20 min (total 1200 pulses) Sham group: Similar parameters to high frequency stimulation producing the same noise, but with the coil rotated away from scalp
Momosaki et al. (2014) [49] Japan	OD as per patient reports of swallowing difficulties. Inclusion: cerebral infarction >6 months ago; mild dysphagia; ≥20 years of age Exclusion: contraindications to magnetic stimulation; cognitive impairment; major general health problems; malignant tumours; skin disease of the neck; carotid vein thrombosis	*n* = 20 Treatment group (10), 50% Functional magnetic stimulation Sham group (10), 50% Sham Functional magnetic stimulation	Treatment group: Age 61 ± 22 80% male Duration post-stroke 19 +/− 8 months Lesion: cerebrum 2, cerebellum 2, brainstem 5, mixed 1 Sham group: Age 66 ± 9 60% male Duration post-stroke 21 +/− 8 months Lesion: cerebrum 1, cerebellum 3, brainstem 2, mixed 4. NS difference between groups.	Procedure: Single session of Functional Magnetic Stimulation or sham using MagVenture MagProR30 (MagVenture Company) Stimulation strength was set at 90% of the minimal intensity at which the patient could subjectively feel local pain High-frequency stimulation of 30 Hz directly to the suprahyoid muscle group, 1200 pulses in total with 10 min in duration. Location of stimulation site unreported. Suprahyoid muscle group defined as being at the midpoint of the hyoid bone and the chin. Sham group: Same parameters with the coil held on its lateral side
Park et al. (2013) [50] Korea	OD as per VFSS Inclusion: >1 month post-stroke, Exclusion: metal implants, pacemaker, history of seizures	*n* = 18 Treatment group (9), 50% rTMS Sham group (9), 50% Sham rTMS	Treatment group: Age 73.7 ± 3.8 56% male Infarct = 7, haemorrhage = 2Right lesion = 6 Sham group: Age 68.9 ± 9.354% male Infarct = 8, haemorrhage = 1Right lesion = 5 NS difference between groups.	Procedure: rTMS (Magstim Rapid2) **Stimulation via Magstim coil positioned over pharyngeal hotspot of intact hemisphere. Stimulation site identified by EMG**. **10 min/day, daily for 2 weeks** Treatment group: Pharyngeal motor thresholds calculated 10 trains of 5-Hz stim, each 10 s, repeated every minute Sham group: Same rTMS dosage, however Magstim coil positioned at 90 degree tilt (same noise, no motor cortical stimulation)
Park et al. (2017) [35] Korea	OD as per VFSS. Inclusion: subacute stroke (unilateral ischemic or haemorrhagic) <3 months post-stroke; swallowing problems lasting >2 weeks; aspiration and/or penetration on VFSS Exclusion: dysphagia from other underlying neurological diseases; history of intractable seizure; metallic implants in the brain	*n* = 33 Treatment group 1 (11), 33.3% Bilateral rTMS Treatment group 2 (11), 33.3% Unilateral rTMS Sham group (11), 33.3% Sham rTMS	Treatment group 1: Age 60.2 ± 13.8 73% male Infarct = 7, haemorrhage = 4 Treatment group 2: Age 67.5 ± 13.4 73% male Infarct = 9, haemorrhage = 2 Sham group: 69.6 ± 8.6 64% male Infarct = 7, haemorrhage = 4 NS difference between groups.	Procedure: rTMS (Magstim Rapid 2) to cortical representation of the mylohyoid muscle, identified by EMG. Applied 10 Hz and 90% of RMT for 5 s with a 55 s inter-train interval. 10 consecutive rTMS sessions. DT for 30 min each day after rTMS Treatment group 1 (Bilateral rTMS): rTMS applied at the ipsilesional motor cortex over the mylohyoid hotspot rTMS applied (same area) to contralesional hemisphere. DT Treatment group 2 (Unilateral rTMS): rTMS applied at the ipsilesional motor cortex over the mylohyoid hotspot Sham rTMS over the contralesional hemisphere DT Treatment group 3 (Sham): Sham rTMS was performed with the coil held at 90° to the scalp, with same stimulation (duration, time, intensity, and frequency) to both hemispheres DT DT included oral sensory training, oral and pharyngeal muscle exercise training, and compensatory techniques.
Tarameshlu et al. (2019) [33] Iran	OD as per Mann Assessment of Swallowing Ability (MASA) Inclusion: >18 years; first-ever stroke; dysphagia >1 month post-stroke Exclusion: presence of dementia; other neurological diseases; history of recurrent stroke; severe aphasia; severe agitation/unconscious	*n* = 18 Treatment group 1 (6), 33.3% rTMS Treatment group 2 (6), 33.3% DT only Treatment group 3 (6), 33.3% DT + rTMS	Treatment group 1: Age 55.33 ± 19.55 67% male 67% cortical stroke, 33% subcortical Treatment group 2: Age 74.67 ± 5.92 17% male 83% cortical stroke, 17% subcortical Treatment group 3: Age 66 ± 5.55 67% male 67% cortical stroke, 33% subcortical NS difference between groups.	Treatment group 1: rTMS (Magstim super-rapid stimulator). Stimulation to intact hemisphere (cortical area for mylohyoid muscles), identified by EMG. Train of 1200 pulses at 1 Hz, stimulus strength at 20% above resting motor threshold. 20 min daily × 5 consecutive days Treatment group 2: Standard swallow therapy (DT). Postural changes (chin up, chin down, head tilt, and head rotation), oral motor exercises, swallowing manoeuvers, and strategies to sensory stimulation alerting volume and speed of food presentation, alerting food consistency and viscosity, and downward pressure of the spoon against the tongue 18 sessions (3 × week) Treatment group 3: Combined rTMS + DT 5 consecutive days rTMS + 18 sessions DT
Ünlüer et al. (2019) [51] Turkey	OD as per VFSS Inclusion: unilateral hemispheric stroke, chronic (2–6 months) oropharyngeal dysphagia, no prior dysphagia rehabilitation and/or cortical stimulation therapy Exclusion: previous dysphagia, other neurogenic disease, epilepsy, tumour, head/neck radiotherapy, unstable medical condition, severe cognitive impairment, severe aphasia, contraindication to magnetic or electrical stimulation	*n* = 28 Treatment group (15), 53.6% rTMS Sham group (13), 46.4% Sham rTMS	Treatment group: Age 67.80 ± 11.88 60% male 7% haemorrhage, 93% ischaemic stroke Sham group: Age 69.31 ± 12.89 46% male 8% haemorrhage, 92% ischaemic stroke NS difference between groups.	Procedure: DT for 30–45 min, 3 days/week (+2 days home exercises) for 4 weeks DT included oropharyngeal muscle strengthening exercises, thermal tactile, stimulation, Masako and Mendelson manoeuvers, vocal fold exercises, Shaker exercises, and tongue retraction exercises Treatment group: DT Combined rTMS (via MMC-140, 33 kT/s, figure 8 coil) delivered in the final 4th week 20 min daily, 5 consecutive days rTMS, 1 Hz (at 90% of threshold intensity) applied to the mylohyoid cortical area of the unaffected hemisphere, identified by EMG. Control group: DT as per above No rTMS delivered in the 4th final week
**transcranial Direct Current Stimulation (tDCS)—*n* = 9**				
Ahn et al. (2017) [36] Korea	OD as per clinical assessment, confirmed by VFSS pre-treatment Inclusion: 18–80 years; first stroke, unilateral (sub)cortical lesion, >6 months ago; able to receive dysphagia therapy 5x a week; no history of abnormal response to brain or electrical stimulation Exclusion: pre-existing major neurological or psychiatric disease; dementia; other brain lesions; risk factors for transcranial direct current stimulation (tDCS)	*n* = 26 Treatment group (13), 50% tDCS + DT Sham group (13), 50% sham-tDCS + DT	Treatment group: Age 61.6 ± 10.3 69.2% male 38.5% infarction, 61.5% haemorrhage Sham group: Age 66.4 ± 10.7 46.2% male 84.6% infarction, 15.4% haemorrhage Statistical difference between groups = NR	Procedure: Bihemispheric anodal tDCS 1 mA stimulation (via Neuroconn GmbH), and standard swallow therapy (DT). 2 anodal electrodes bilaterally to the pharyngeal motor cortices (site location method not described). 2 cathodal references electrodes attached to both supraorbital regions of the contralateral hemisphere. DT included compensatory methods, behavioural manoeuvres, oromotor exercises and thermal tactile stimulation Treatment group + DT: Ten 20 min sessions (5 times a week for 2 weeks) Sham + DT: 30 s through 2 anodal electrodes–tingling sensation, but no changes in cortical excitability
Cosentino et al. (2020) [31] Italy	OD as per clinical assessment and FEES. Inclusion: presbydysphagia for ≥6 months due to Central ervous System disorder; ≥65 years Exclusion: unstable medical condition; cognitive impairment; severe dysphagia with inability to swallow liquid or semiliquid boluses; contraindications to stimulation used in study	*n* = 40 Treatment group 1 (17), 42.5% tDCS Treatment group 2 (23), 57.5% Theta-burst stimulation (TBS) Both groups crossed over to sham treatment, also. Order randomised.	Treatment group 1: Age 71.5 ± 5.2 53% male 70.5% primary presbydysphagia, 72.4% secondary presbydysphagia Treatment group 2: Age 75.2 ± 4.8 (*p* = 0.025) 57% male 76.4% primary presbydysphagia, 74.0% secondary presbydysphagia Statistical difference between groups = NR	Procedure: tDCS or TBS (Transcranial Magnetic Stimulation Unit STM9000, Ates Medica Device) 5 sessions over 5 consecutive days Anode electrode placed over the right swallowing motor cortex; cathode positioned over the contralateral orbitofrontal cortex. Optimal location identified as the site where 3/5 consecutive, low intensity magnetic stimuli elicited MEPs of minimum 50 microV from resting contralateral submental muscles complex. Treatment group 1: tDCS at 1.5 mA (ramped up or down for the first and last 30 s) over 20 min Sham treatment similar for patient, DC stimulator turned off after 30 s of stimulation Treatment group 2: TBS: three 50 Hz magnetic pulses repeated every 200 ms for 2 s. Each cycle repeated every 10 s for 20 times (600 pulses in total) Sham treatment parameters set as for real TBS with coil positioned at 90 degrees against the skull
Kumar et al. (2011) [27] USA	OD as per Dysphagia Outcome and Severity Scale (DOSS) score by SLT. In cases of ambiguity about appropriate DOSS score, VFSS was performed (required with 7 patients) Inclusion: first ischaemic stroke 24–168 h ago, new onset dysphagia with DOSS score ≤5 Exclusion: difficulty following instructions, pre-existing swallowing problems, contraindications to anodal transcranial direct current stimulation (tDCS)	*n* = 14 (pilot study) Treatment group (7), 50% tDCS + DT Sham group (7), 50% Sham tDCS + DT	Treatment group: Average age 79.7 43% male Average NIHHS score 13.6Sham group: Average age 70 57% male Average NIHHS score 13.1Statistical difference between groups = NR	Procedure: tDCS or sham (via Phoresor; Iomed stimulator) 5 consecutive days (2 mA for 30 min to the nonlesional hemisphere). Site location identified by MRI or CT. Electrode placed over the undamaged hemisphere, mid-distance between C3-T3 on left, or C4-T4 on right; reference electrode over the contralateral supraorbital region Concurrent DT–patients sucked on a lemon-flavoured lollipop doing effortful swallows (~60x each session) Sham group + DT: Treatment parameters not described in detail
Pingue et al. (2018) [37] Italy	OD as per clinical swallow examination and DOSS <5 Inclusion: unilateral stroke < 4 weeks prior to enrolment; age > 18 years; no other muscular or neurological disease or severe disorder of consciousness; mild to severe dysphagia (DOSS < 5); National Institutes of Health Stroke Scale (NIHSS) <22 Exclusion: history of dysphagia, other severe clinical conditions (eg, severe infections), potential contraindications to tDCS	*n* = 40 Treatment group (20), 50% tDCS + DT Sham group (20), 50% Sham tDCS + DT	Treatment group: Age 63.5 (range = 54.5–75.25) 40% male Infarct = 11, haemorrhage = 11 (NB. Note numeral errors reported here, *n*= 20, not 22) Sham group: Age 68.5 (range = 62–73) 40% male Infarct = 4, haemorrhage = 16 NS difference between groups.	Procedure: tDCS by a battery-driven constant current stimulator (HDCkit Newronika, Italy). Stimulation targeted the pharyngeal motor cortex (site location method not described). 30 min stimulation was applied during swallowing rehabilitation 10 sessions over 10 days DT: Direct = compensatory methods, behavioural manoeuvers, supraglottic and effortful swallowing). Indirect approaches = physical manoeuvers, thermal tactile stimulation. Treatment group + DT: 2 mA of anodal tDCS over the lesioned hemisphere and cathodal stimulation to the contralesional hemisphere. Sham + DT: Same protocol except current was delivered for only 30 s through 2 electrodes, producing initial tingling sensation but no cortical excitability.
Sawan et al. (2020) [29] Egypt	OD as per bedside swallow assessment as pre-treatment VFSS Inclusion: acute or subacute carotid system ischaemic stroke; stable, oriented and able to follow commands; presence of dysphagia Exclusion: pre-existing severe dysphagia; difficulty communicating; impaired cognition; neuro-degenerative disorder; major psychiatric illness; unstable health issues such as severe cardiac disease or renal failure; intracranial devices and/or metal; pacemaker or other implanted electrically sensitive device; chronic drug use that could affect brain activity; epilepsy; pregnancy	*n* = 40 Treatment group (20), 50% tDCS + DT (physical therapy program) Sham group (20), 50% Sham tDCS + DT (physical therapy program)	Treatment group: Age 53.3 ± 5.0 50% unilateral stroke, 50% bilateral stroke Sham group: Age 50.3 ± 5.2 50% unilateral stroke, 50% bilateral stroke NS difference between groups.	Procedure: tDCS or Sham Stimulation targeted the pharyngeal motor cortex (site location method not described), using neuromodulation technology (Soterix medical Inc., New York, NY, USA). 30 min stimulation with a constant current of 2 mA intensity 5 consecutive sessions for 2 weeks Physical therapy program to improve swallowing: details NR Treatment group + DT: Group 1 (unilateral hemispheric stroke) anode placed on healthy hemisphere with reference electrode over contralateral supraorbital region.Group 2 (bilateral hemispheric stroke) stimulation first applied to the dominant hemisphere, then non-dominant hemisphere. Sham + DT: Same protocol producing tingling sensation but no cortical excitability.
Shigematsu et al. (2013) [38] Japan	Severe OD as per clinical swallow examination, confirmed by VFSS and FEES, tube-feeding. Inclusion: ≥4 weeks post-stroke; admitted to rehabilitation hospital Exclusion: subarachnoid haemorrhage; history of seizures; severe consciousness disturbance; organic neck disease; history of surgery; no other muscular or neurological disorders.	*n* = 20 Treatment group (10), 50% tDCS + DT Sham group (10), 50% Sham tDCS + DT	Treatment group: Age: 66.9 ± 6.3 70% male; Time post-stroke: 12.9 ± 7.8 Site of lesion: 20% putamen; 20% medulla oblongata; 10% corona radiata; 10% frontotemporal; 10% frontoparietal; 10% pons; 10% thalamus; 10% internal capsule Sham group: Age 64.7 ± 8.9 70% male Time post-stroke: 12.1 ± 9.0 Site of lesion: 40% pons; 20% frontoparietal; 10% putamen; 10% thalamus; 10% internal capsule; 10% caudate nucleus NS difference between groups.	Procedure: stimulation by DC stimulator (NeuroConn) 1-mA anodal tDCS to ipsilateral pharyngeal motor cortex area, cathode placed contralesional supraorbital region (site location method not described). 1 × day, 10 days (2 × blocks of 5 days) Simultaneous intensive DT (based on VFSS and FEES) including thermal-tactile stimulation, supraglottic swallow, effortful swallow, Shaker exercise, K-point stimulation, blowing Treatment group + DT: 20 min tDCS with simultaneous intensive DT Sham + DT: Same stimulation set-up, for 40 s only Same intensive DT.
Suntrup-Krueger et al. (2018) [39] Germany	OD as per FEES Inclusion: dysphagia due to acute ischemic stroke, confirmed by brain imaging; >18 years; >24 h post-stroke onset Exclusion: pre-existing swallowing difficulties, contraindications to tDCS, tracheal cannula, unstable medical condition, inability to stay alert	*n* = 59 Treatment group (29), 49.2% tDCS + DT Sham group (30), 50.8% Sham tDCS + DT	Treatment group: Age 68.9 ± 11.5 58.6 % male 72.4% supratentorial stroke 27.6% infratentorial stroke Sham group: Age 67.2 ± 14.5 56.7 % male 80.0% supratentorial stroke 20.0% infratentorial stroke NS difference between groups.	Procedure: tDCS stimulation delivered by battery-driven constant current stimulator (NeuroConn) 1 mA anodal tDCS, 1 × day, 4 consecutive days. Stimulation to area of the motor cortical swallowing network of intact hemisphere in cortical stroke patients; stimulation applied to right hemisphere in brainstem stroke patients (site location method not described, but rationale provided). Swallow exercises performed during stimulation, if appropriate. Treatment group + DT: Anodal tDCS for 20 min with simultaneous swallow exercises, if appropriate Sham + DT: Stimulation for 30 s only, with electrodes left in place for 20 min
Wang et al. (2020) [40] China	OD caused by cricopharyngeal muscle dysfunction as per VFSS Inclusion: brainstroke with cricopharyngeal muscle dysfunction, onset duration >1 month prior to enrolment; aspiration as per VFSS; nasogastric tube in-situ; MMSE ≥ 23 Exclusion: severely decreased consciousness; history of epilepsy; unstable medical condition; history of previous dysphagia; history of radiotherapy for head and neck diseases; intracranial metallic device	*n* = 28 Treatment group (14), 50% tDCS + DT Sham group (14), 50% Sham tDCS + DT	Treatment group: Age 61.43 ± 11.24 79% male Time post-stroke: 66.79 ± 38.62 days Sham group: Age 62.00 ± 10.46 71% male Time post-stroke: 67.50 ± 47.62 days NS difference between groups.	Procedure: anodal tDCS + catheter balloon dilatation + standard swallow therapy (based on VFSS, details not described) 20 sessions, 5 × week for 4 weeks Treatment group + DT + Balloon dilatation: tDCS via IS300 (Zhineng Electronics Industrial Co.) 1 mA anodal stimulation to oesophageal cortical areas, bilaterally (site location method not described). Each hemisphere stimulated for 20 min (interval of 30 min between). Combined with catheter balloon dilation and standard swallow therapy Sham + DT + Baloon dilatation: tDCS anodal stimulation as per treatment except for 30 s only Combined with catheter balloon dilation and standard swallow therapy
Yang et al. (2012) [41] South Korea	OD as per clinical swallow examination (VFSS at baseline) Inclusion: first ever ischemic stroke ≤2 months ago; use of a nasogastric tube Exclusion: bilateral brain lesion; tDCS contraindicators; unstable medical condition; severe language disturbance; neglect, depression, or cognitive deficits (MMSE, ≤10/30 points); history of severe alcohol or drug abuse; taking Na+ or Ca2+ channel blockers or N-methyl-D-aspartate (NMDA) receptor antagonists; previous stroke that resulted in residual disability	*n* = 16 Treatment group (9), 56.3% tDCS + DT Sham group (7), 43.7% Sham tDCS + DT	Treatment group: Age 70.44 ± 12.59 66.7% male 44.4% right lesion, NIHSS = 9.7 ± 5.4 Sham group: Age 70.57 ± 8.46 42.9% male 57.1% right lesion, NIHSS = 13.9 ± 6.3 NS differences between groups.	Procedure: anodal tDCS (Phoresor II) Stimulation, at 1 mA, to the pharyngeal area of the affected hemisphere (site location method not described). 5 times/week for 2 weeks DT = diet modifications, positioning, Mendelsohns manoeuver, supraglottic, effortful swallowing, thermal tactile stimulation and oral motor exercises Treatment group + DT: 20 min tDCS + simultaneous DT DT alone, continued for another 10 min Sham + DT: 30 s tDCS + simultaneous DT DT alone, continued for another 10 min
**Combined Neurostimulation Interventions—*n* = 4**				
Cabib et al. (2020) [44] Spain	OD as per VFSS Inclusion: > 3 months post-unilateral stroke, stable medical condition Exclusion: neurodegenerative disorders, epilepsy, drug dependency, brain or head trauma or surgery, structural causes of OD, pacemaker or metallic body implants, and pregnancy or lactation	*n* = 36 Treatment group 1 (12), 33.3% rTMS Treatment group 2 (12), 33.3% Capsaicin Treatment group 3 (12), 33.3% PES	Treatment group 1: Age 70.0 ± 8.6 75% male 0% haemorrhage, 100% infarction Treatment group 2: Age 74.3 ± 7.8 58% male 8% haemorrhage, 92% infarction Treatment group 3: Age 70.0 ± 14.2 92% male 25% haemorrhage, 75% infarction NS differences between groups, except shorter time since stroke for capsaicin group.	Procedure: All patients received both treatment and sham, cross-over active/sham in visits 1 week apart (randomised). Assessment occurred immediately prior to treatment and within 2 h post-treatment.Treatment group 1: rTMS (Magstim rapid stimulator) Stimulation (90% of threshold) bilaterally to motor hotspots for pharyngeal cortices, identified by EMG. 5 Hz train of 50 pulses for 10 s × 5 (total 250 pulses), 10 s between trains Sham = coil tilted 90 degrees Treatment group 2: Capsaicin stimulus (10^−5^ M) or placebo (potassium sorbate) were administered once in a 100 mL solution Treatment group 3: PES via two-ring electrode naso-pharyngeal catheter (Gaeltec Ltd.) 10 min stimulation at 75% tolerance threshold (0.2 ms of duration) and 5 Hz Sham = 30 s of above stimulation then no stimulation
Lim et al. (2014) [43] Korea	OD as per VFSS Inclusion: primary diagnosis unilateral cerebral infarction or haemorrhage (CT or MRI); stroke onset <3 months; patients who could maintain balance during evaluation + treatment; and adequate cognitive function to participate Exclusion: could not complete VFSS/failed the examination; presence of dysphagia pre-stroke; history of prior stroke, epilepsy, tumor, radiotherapy in the head and neck, or other neurological diseases; unstable medical condition; and contraindication to magnetic or electrical stimulation	*n* = 47 Treatment group 1 (15), 31.9% DT Treatment group 2 (14), 29.8% DT + rTMS Treatment group 3 (18), 38.3% DT + NMES	Treatment group 1: Age 62.5 ± 8.2 60% male 34% haemorrhage, 66% infarction Treatment group 2: Age 59.8 ± 11.8 43% male 71% haemorrhage, 29% infarction Treatment group 3: Age 66.3 ± 15.4 67% male 66% haemorrhage, 44% infarction NS difference between groups.	Procedure: DT: oropharyngeal muscle-strengthening, exercise for range of motion of the neck/tongue, thermal tactile stimulation, Mendelson maneuver, and food intake training for 4 weeks Treatment group 1: DT 4 weeks Intensity NR Treatment group 2: DT + rTMS via Magstim 200 (Magstim, Whiteland, UK) Stimulation to pharyngeal motor cortex, contralateral hemisphere; optimal stimulation site located by EMG. 1 Hz stimulation, 100% intensity of resting motor threshold 20 min/day, (total 1200 pulses a day), 5 × week for 2 weeks Treatment group 3: DT + NMES (Vitalstim) 300 ms, 80 Hz (100 ms in interstimulus intervals). Intensity between 7–9 mA, depending on patient compliance. Stimulation to supra and infra hyoid region 30 min/day, 5 days/week, 2 weeks
Michou et al. (2014) [42] UK	OD as per diagnoses made by SLT (confirmed with VFSS at start of treatment) Inclusion: post-stroke dysphagia for >6 weeks Exclusion: Hx of dementia, cognitive impairment, epilepsy, head and neck surgery; neurological defects prior to stroke; cardiac pacemaker or defibrillator in-situ; severe concomitant medical conditions; structural oropharyngeal pathology; intracranial metal; pregnancy; medications acting on Central Nervous System.	*n* = 18 Treatment group 1 (6), 33.3% Pharyngeal electrical stimulation (PES) Treatment group 2 (6), 33.3% Paired associative stimulation (PAS) Treatment group 3 (6), 33.3% rTMS	Treatment group: Avg age 60.3 83% male Treatment group 2: Avg age 67.3 100% male Treatment group 3: Avg age 67.8 66.7% male Overall 63 +/− 15 weeks post-stroke with 7.6 +/− 1 on NIHHS Statistical difference between groups = NR	Procedure: Single application of neurostimulation using a figure of 8 shaped magnetic coil connected to a Magstim BiStim2 magnetic stimulator (Magstim Co, UK) All patients received real and sham treatment in randomised order on two different days Treatment group 1: PES Frequency of 5 Hz for 10 min. Intensity set at 75% of the difference between perception and tolerance thresholds. Treatment group 2: Paired associative stimulation: Pairing a pharyngeal electrical stimulus (0.2 ms pulse) with a single TMS pulse over the pharyngeal MI at MT intensity plus 20% of the stimulator output. The 2 pulses were delivered repeatedly every 20 s with an inter-stimulus interval of 100 ms for 10 min. Treatment group 3: rTMS Stimuli to pharyngeal motor cortex, identified by EMG, with the TMS coil. Frequency of 5 Hz, intensity 90% of resting thenar motor threshold in train of 250 pulses, in 5 blocks of 50 with 10 s between-blocks pause.
Zhang et al. (2019) [45] China	OD as per DOSS by a well-trained doctor Inclusion: stroke as per MRI <2 months earlier; aged 50–75 years; normal consciousness, stable vital signs, presence of dysdipsia and dysphagia Exclusion: brain trauma or other central nervous system disease; unstable arrhythmia, fever, infection, epilepsy, or use of sedative drugs; poor cooperation due to serious aphasia or cognitive disorders; contraindications to magnetic or electrical stimulation	*n* = 64 Treatment group 1 (16), 25.0%. Sham rTMS + NMES Treatment group 2 (16), 25.0% Ipsilateral rTMS + NMES Treatment group 3 (16), 25.0% Contralateral rTMS + NMES Treatment group 4 (16), 25.0% Bilateral rTMS + NMES	Treatment group 1: Age 55.9 ± 8.9 43% male 61.5% subcortical, 38.5% brainstem Treatment group 2: Age 56.8 ± 9.7 54% male 30.8% subcortical, 69.2% brainstem Treatment group 3: Age 56.5 ± 10.1 50% male 58.3% subcortical, 41.7% brainstem Treatment group 4: Age 53.1 ± 10.6 31% male 61.5% subcortical, 38.5% brainstem All data given on participants that finished the trial and follow-up period (*n* = 52)	Procedure: 10 rTMS (sham or real) and 10 NMES sessions Mon-Fri during 2 weeks NMES: 30 min once daily using a battery powered handheld device (HL-08178B; Changsha Huali Biotechnology Co., Ltd., Changsha, China), vertical placement of electrodes. Pulse width of 700 ms, frequency 30–80 Hz, current intensity 7–10 mA. rTMS delivered by figure-of-eight coil (CCY-IV; YIRUIDE Inc., Wuhan, China) during NMES with a sequence of HF-rTMS over the affected hemisphere followed by LF-rTMS over the unaffected hemisphere (site location method not described). HF-rTMS parameterss: 10 Hz, 3 s-s stimulation, 27 s-s interval, 15 min, 900 pulses, and 110% intensity of resting motor threshold (rMT) at the hot spot LF-rTMS parameters: 1 Hz, total of 15 min, 900 pulses, and 80% intensity of rMT at the hot spot Treatment group 1: Sham rTMS + NMES 10-Hz sham rTMS delivered to the hot spot for the mylohyoid muscle at the ipsilesional hemisphere followed by 1-Hz sham rTMS over the corresponding position of the contralesional hemisphere Delivered using a vertical coil tilt, generating the same noise as real rTMS without cortical stimulation Treatment group 2: Ipsilateral rTMS + NMES 10-Hz real rTMS was delivered to the hot spot for the mylohyoid muscle at the ipsilesional hemisphere followed by 1-Hz sham rTMS over the corresponding position of the contralesional hemisphere. Treatment group 3: Contralateral rTMS + NMES 10-Hz sham rTMS was delivered to the hot spot for the mylohyoid muscle at the ipsilesional hemisphere followed by 1-Hz real rTMS over the corresponding position of the contralesional hemisphere. Treatment group 4: Bilateral rTMS + NMES 10-Hz real rTMS was delivered to the hot spot for the mylohyoid muscle at the ipsilesional hemisphere followed by 1-Hz real rTMS over the corresponding position of the contralesional hemisphere.

^a^ Where information was available on how stimulation site was located and mapped, and whether stimulation was applied ipsilateral or contralateral to the lesion site, it was included. Note. NMES is at motor stimulation level unless explicitly mentioned. Notes. CP—cerebral palsy; CT—computed tomography; DOSS—dysphagia outcome and severity scale; DT—dysphagia therapy; EMG—electromyography; FEES—fiberoptic endoscopic evaluation of swallowing; FOIS—functional oral intake scale; MEP—motor-evoked potentials; MMSE—Mini-Mental State Exam; MRI—magnetic resonance imaging; MS—multiple sclerosis; MT—Motor Threshold; NIHSS—National Institutes of Health Stroke Scale; NMES—neuromuscular electrical stimulation; NR—not reported; NS—not significant; OD—oropharyngeal dysphagia; OST—oral sensorimotor treatment; PAS—penetration—aspiration scale; PES—pharyngeal electrical stimulation; rTMS—repetitive transcranial magnetic stimulation; SLT—Speech and Language Therapist; TBI—traumatic brain injury; tDCS—transcranial direct current stimulation; TOR-BSST—Toronto Bedside Swallowing Screening test; VFSS—videofluoroscopic swallowing study.

**Table 3 jcm-11-00993-t003:** Outcome of rTMS and tDCS for people with oropharyngeal dysphagia.

Study	Intervention Goal	Outcome Measures	Intervention Outcomes &Conclusions
**repetitive Transcranial Magnetic Stimulation (rTMS)—*n* = 11**			
Cheng et al. (2017) [46]	To investigate the short-(2-months) and long-term (6 and 12 months) effects of 5 Hz rTMS on chronic post-stroke dysphagia	Primary outcomes: Maximum tongue strength, VFSS (oral transit time, stage transit time, pharyngeal transit time, pharyngeal constriction ratio), and SAPP [52]. Assessed: 1 week pre-, and 2, 6 and 12 months post-intervention.	No significant differences between groups at any time point post-treatment for any of the VFSS measures nor for tongue strength No significant different between groups for the SAPP outcome measure
Du et al. (2016) [34]	To investigate the effects of high-frequency versus low-frequency rTMS on poststroke dysphagia during early rehabilitation	Primary outcome: SSA [53]. Secondary outcomes: WST [54], DD [55], NIHSS score [56], BI [57], mRS, measures of mylohyoid MEPs evoked from both hemispheres before and after treatment. Assessed: before treatment, after 5th rTMS session, and at 1-, 2-, and 3-months post-treatment.	Primary outcomes: SSA scores improved in both 3 Hz and 1 Hz rTMS groups and maintained over 3-month follow-up (*p* = 0.001, compared to Sham) Secondary outcomes: *Both treatment groups compared to sham.* WST scores significantly better at 5 days (*p* = 0.017), 1 month (*p* = 0.002), 2 months (*p* < 0.001) and 3 months (*p* < 0.001) DD scores significantly improved at 1 month (*p* = 0.001), 2 months (*p* < 0.001) and 3 months (*p* < 0.001) BI and mRS improved in all patients 1 Hz rTMS induced a decrease in the cortical excitability of the unaffected hemisphere, but an increase in that of the affected hemisphere 3 Hz rTMS enhanced the cortical excitability of the affected hemisphere and slightly affected that of the unaffected hemisphere
Khedr et al. (2009) [47]	To investigate the therapeutic effect of rTMS on post-stroke dysphagia	Primary outcome: Dysphagia rating scale [58] (swallowing questionnaire + bedside examination). Secondary outcomes: Motor power of hand grip, BI [57], measures of oesophageal MEPs from both hemispheres. Assessed: before and immediately after treatment, and at 1- and 2-months post-treatment.	Dysphagia scores significantly better in the treatment group (no *p* value or CI given), maintained at 2 months post Hand grip strength and BI improved in both groups. Improvement in BI greater in the treatment group Conclusion: rTMS led to a significantly greater improvement in dysphagia and motor disability that was maintained at 2 months.
Khedr and Abo-Elfetoh (2010) [48]	To assess the effect of rTMS on dysphagia in patients with acute lateral medullary or other brainstem infarction	Primary outcome: DD [55] Secondary outcomes: Hand grip strength, NIHHS [56] and BI [57]. Assessed: before treatment, after 5th rTMS session, and at 1- and 2-months post-treatment.	Results given based on infarction type divided into treatment versus sham. rTMS and lateral medullary infarction Significant improvement in DD in the treatment group when compared to sham Barthel Index improved significantly more in the treatment group compared to sham. No significant difference in other secondary outcomes between groups. Hand grip strength and NIHHS improved in both groups rTMS and other brainstem infarction Significant improvement in DD in the treatment group when compared to sham No significant difference in secondary outcomes between groups
Khedr et al. (2019) [30]	To investigate the therapeutic effect of rTMS on dysphagia with Parkinson’s Disease	Primary outcomes: Hoen and Yahr staging [59], UPDRS [60] part III, IADL [61], Self-Assessment Scale [62], SDQ [63], Arabic-DHI [64]. VFSS was conducted on 9 rTMS and 6 sham group patients. Assessed: before treatment, post treatment, and at 1-, 2-, and 3-months post-treatment.	Mean change in UPDRS III was significantly higher in the treatment group (*p* = 0.0001) Mean reduction in the Arabic-DHI was significantly greater in the treatment group (*p* = 0.0001) VFSS (*n* = 15): significant improvement in hyoid bone excursion and pharyngeal transit time for fluid swallows in the treatment group (*p* = 0.04 and 0.03 respectively). No difference in AP scores or residue. Results for IADL, SDQ or self-assessment scale = NR Conclusion: rTMS improves dysphagia in PD
Kim et al. (2011) [32]	To investigate the effect of rTMS on dysphagia recovery in patients with brain injury	Primary outcomes: FDS [65], PAS [66] and ASHA-NOMS [67] before and after treatment Assessed: before and after treatment, times unspecified.	FDS and PAS improved significantly in the low intensity group compared to other groups Significant improvement in ASHA-NOMS Swallow Scale in the sham and low intensity groups
Momosaki et al. (2014) [49]	To assess the effectiveness of a single functional magnetic stimulation session on post-stroke dysphagia	Primary outcomes: Timed WST [54] before and after stimulation Secondary outcome: N/R	Significant improvement in speed (*p* = 0.008) and capacity (*p* = 0.005) for the treatment group compared to sham No significant differences in inter-swallow interval between groups. Within group changes not reported.
Park et al. (2013) [50]	To find the therapeutic effect of high-frequency repetitive TMS on a contra-lesional intact pharyngeal motor cortex inpost-stroke dysphagic patients	Primary outcome: VDS [68], PAS [66] (as per VFSS), pre- and post- treatment. 2 and 4 weeks from baseline. Secondary outcomes: Oral and pharyngeal components of VDS	Treatment group: Significantly improved (*p* > 0.05) VDS scores post-treatment at 2 and 4 weeks. Significantly improved (*p* > 0.05) PAS at 2 and 4 weeks Sham: NS difference between pre-post measures at 2 or 4 weeks for either VDS or PAS measures
Park et al. (2017) [35]	to investigate the effects of high-frequency rTMS at the bilateral motor cortices over the cortical representation of the mylohyoid muscles in the patients with post-stroke dysphagia.	Primary outcomes: Immediately post-treatment and 3 weeks post-treatment: using CDS [69], DOSS [58], PAS [66], and VDS [68]. Secondary outcome: N/R	Significant difference (*p* < 0.05) was found between the bilateral rTMS versus unilateral rTMS and Sham groups across all time-points post-treatment Bilateral treatment group1: CDS, DOSS, PAS and VDS improved significantly (*p* > 0.05) post-treatment + 3 weeks post-treatment Unilalteratl treatment group2: CDS, DOSS, PAS and VDS improved significantly (*p* > 0.05) post-treatment + 3 weeks post-treatment Sham: DOSS, PAS and VDS improved significantly post-treatment and 3 weeks post-treatment. CDS improved immediately post-treatment only
Tarameshlu et al. (2019) [33]	To compare the effects of standard swallow therapy (DT), rTMS and a combined intervention (CI)on swallowing function in patients with poststroke dysphagia	Primary outcome: MASA [70]. Secondary outcomes: FOIS [71] assessed (a) before treatment, (b) after 5th session and after 10th, 15th and 18th session.	Primary outcome: MASA No significant difference between groups after 5th treatment session After 18 sessions: no significant difference between Treatment group 1 and Treatment group 2, nor Treatment group 2 and Treatment group 3 Significant difference (*p* = 0.01) between Treatment group 3 which improved greater than Treatment group 1 Secondary outcomes: FOIS No significant difference between groups after 5th treatment session After 10 and 18 sessions: no significant difference between Treatment group 1 and Treatment group 2 After 10 and 18 sessions, significant difference between Treatment group 3 (greater improvement) versus Treatment group 1 (*p* = 0.03 and 0.004 for 10 and 18 sessions respectively) and sigficiant improvement for Treatment group 3 versus group 2 (*p* = 0.01 and 0.02 for 10 and 18 months respectively) All groups showed within-group improvements.
Ünlüer et al. (2019) [51]	To identify whether applying low-frequency rTMS can enhance the effect of conventional swallowing treatment and quality of life of chronic (2–6 months) stroke patients suffering from dysphagia	Primary outcome: PAS [66], pre-post treatment, 1 and 3 months post-treatment. Secondary outcomes: VFSS parameters (including oral parameters, tongue retraction, hyolaryngeal elevation, delayed swallow reflex, residue, nutritional status, SWAL-QOL).	No significant difference between groups at 1 and 3 months post-treatment across any of the outcome measures Treatment group PAS scores improved (*p* = 0.035) for liquid swallows only at the 1 month post-treatment assessment Control group PAS scores (for liquids and semi-solids) improved statistically (*p* < 0.05) from baseline to 1 and 3 months post-treatment Variable improvements in secondary outcome measures across both treatment and control group at different time-points.
**transcranial Direct Current Stimulation (tDCS)—*n* = 9**			
Ahn et al. (2017) [36]	To investigate the effect of bihemispheric anodal tDCS with conventional dysphagia therapy on chronic post-stroke dysphagia	Primary outcome: DOSS [58] score based on VFSS pre- and post-treatment Secondary outcome: N/R	No significant difference in DOSS improvement between groups Significant improvement (*p* = 0.02) of DOSS (from 3.46 pre-Tx to 4.08 post-Tx) in the treatment group Improvement (NS) in the sham group (from 3.08 to 3.46) tDCS combined with conventional swallow therapy was not found to be superior to conventional dysphagia therapy with sham treatment
Cosentino et al. (2020) [31]	To investigate the therapeutic potential of tDCS and theta-burst stimulation on primary or secondary presbydysphagia	Primary outcomes: DOSS [58] based on bedside assessment and FEES. Similarity Index based on Electrokinesiographic/electromyographic Study (EES) for Laryngeal-pharyngeal Mechanogram (LPM) and electromyographic activity of the submental/suprahyoid muscles complex (SHEMG). Secondary outcome: N/R Outcomes assessed at baseline, 1 month and 3 months post-treatment	Both Treatment groups 1 and 2, as well as sham improved post-intervention period Treatment group 1: tDCS significantly improved DOSS at 1 month post-treatment (*p* = 0.014). tDCS at 3 months and sham groups improved, though NS tDCS improved Simlarity Index at 1 month post-treatment (*p* = 0.005 for SHEMG-Similarity Index and *p* = 0.04 for LPM-Similarity Index) Treatment group 2: Theta Burst Stimulation improved DOSS score at 1 month (*p* = 0.001) and 3 months post-treatment (*p* = 0.005). Sham improved = NS. Theta Burst Stimulation improved Similarity Index at 1 month post-treatment only in patients with secondary presbydysphagia (*p* = 0.02)
Kumar et al. (2011) [27]	To investigate whether anodal tDCS in combination with swallowing manoeuvres facilitates dysphagia recovery in stroke patients during early stroke convalescence	Primary outcome: DOSS [58]. Secondary outcome: N/R	Treatment group had significantly improved DOSS scores compared to sham group (*p* = 0.019).
Pingue et al. (2018) [37]	To evaluate whether anodal tDCS over the lesioned hemisphere and cathodal tDCS to the contralateral one during the early stage of rehabilitation can improve poststroke dysphagia	Primary outcome: DOSS [58], PAS [66] post-treatment. Secondary outcome: N/R	No significant difference between groups for DOSS or PAS post-treatment Within group: PAS scores improved for both the treatment and sham groups after 6 weeks
Sawan et al. (2020) [29]	To assess the effect of tDCS on improving dysphagia in stroke patients	Primary outcomes: DOSS [58]; Oral Transit Time; laryngeal and hyoid elevation; oesophageal sphincter spasm; aspiration Secondary outcome: N/R	Significant improvement in all variables when comparing treatment group to sham: DOSS score (*p* < 0.001), oral transit time (*p* = 0.004); laryngeal elevation, hyoid elevation and oesophageal sphincter spasm (all *p* < 0.001) and aspiration (*p* = 0.001) Significant improvement in all variables post-tDCS in the treatment group No significant changes in any variables in the sham group
Shigematsu et al. (2013) [38]	To investigate if the application of tDCS to the ipsilateral cortical motor and sensory pharyngeal areas can improve swallowing function in poststroke patients	Primary outcome: DOSS [58] immediately post-treatment and 1 month post-treatment Secondary outcomes: PAS [66], oral intake status.	Significant difference between groups post-treatment and 1-month post-treatment = not reported. tDCS: improved significantly from baseline to post-treatment (*p* = 0.006), and 1-month post-treatment (*p* = 0.004) in DOSS measures. PAS and oral intake reported descriptively Sham: improved significantly from baseline to 1-month post-treatment (*p* = 0.026)
Suntrup-Krueger et al. (2013) [39]	To evaluate the efficacy of a pathophysiologically reasonable tDCS protocol to improve stroke-related OD, via a randomized controlled trial (RCT) in a sufficiently large patient sample with objective clinical outcome measures alongside functional neuroimaging	Primary outcome: Improved FEDSS 4 days post-treatment Secondary outcomes: DSRS [72]; final FEDSS, and FOIS [71] scores prior to discharge; pneumonia rate until discharge; length of stay (in hospital). Activation changes in the swallowing network as measured with MEG.	Primary outcome: FEDSS = both groups improved, statistically significantly greater improvements with treatment group (*p* < 0.001) Secondary outcomes: DSRS = statistically significantly greater improvements with treatment group (*p* = 0.001) FOIS = statistically significantly greater improvements with treatment group compared to sham, at discharge (*p* = 0.041) No other significant differences between groups for other secondary outcomes.
Wang et al. (2020) [40]	To investigate the effects of tDCS combined with conventional swallowing training on the swallowing function in brainstem stroke patients with cricopharyngeal muscle dysfunction.	Primary outcome: FDS [65] (before and immediately after intervention). Secondary outcomes: FOIS [71], MBSImp [73], PESO measurement [74].	Primary outcomes: Statistical difference between the groups at endpoint not reported. tDCS treatment group improved to a greater extent than the sham group post-treatment for thin fluids (IDDSI-0) and thick fluids (IDDSI-3), *p* < 0.001 and *p* = 0.001, respectively Secondary outcomes: FOIS and PESO scores improved to a statistically greater extent (both thin and thick fluids) for the tDCS group versus sham. FOIS *p* = 0.001; PESO *p* = 0.003.
Yang et al. (2012) [41]	To investigate the effects of anodal tDCS combined with swallowing training for post-stroke dysphagia.	Primary outcome: FDS [65] immediately post-treatment and at 3 months Secondary outcomes: Oral Transit Time, Pharyngeal Transit Time and total transit time.	At 1 month: both tDCS and sham group functional dysphagia scale improved immediately post-treatment. NS between groups. At 3 months: significantly greater FDS improvement (*p* = 0.041) with the tDCS group versus sham group, when adjusted for NIHSS score, baseline FDS score, age, lesion size and time from stroke onset (NB. Between group differences at baseline = NS) Secondary outcomes showed no significant differences between the groups.
**Combined Neurostimulation Interventions—*n* = 4**			
Cabib et al. (2020) [44]	To investigate the effect of rTMS of the primary sensory cortex (A), oral capsaicin (B) and intra-pharyngeal electrical stimulation (IPES; C) on post-stroke dysphagia	Primary outcomes: Effect size pre-post treatment for neurophysiological variables (pharyngeal and thenar RMT and MEP). Secondary outcomes: Effects on the biomechanics of swallow (PAS [66], impaired efficiency + more) VFSS before and after treatment	Between group differences (post-treatment) not reported Primary outcomes: ∙ No significant differences in pre-post pharyngeal RMTs with any of the active or sham conditions∙ Combined analysis (interventions grouped together) showed significantly shorter latency times, increased amplitude, and area of the thenar MEP in the contralesional hemisphereSecondary outcomes: (VFSS) No significant change/difference in effect size across any of the treatment or sham groups
Lim et al. (2014) [43]	To investigate the effect of low-frequency rTMS and NMES on post-stroke dysphagia.	Primary outcomes: VFSS baseline, 2- + 4-weeks post-treatment (for semi-solids and liquids): FDS [65], PAS [66], Pharyngeal Transit Time. Secondary outcome: N/R	Difference between groups post-treatment = NR FDS outcome: For semi-solids all groups improved, no significant difference in pre-post change, between groups For liquids, the rTMS and NMES improved significantly compared to DT, 2 weeks post-treatment (*p* = 0.016 and *p* < 0.001, respectively) No significant difference in the change from baseline to the 4th week evaluation among groups (*p* = 0.233) PAS outcome: For semi-solids all groups improved, no significant difference in pre-post PAS change, between groups For liquids, the rTMS and NMES improved significantly compared to DT, 2 weeks post-treatment (*p* = 0.011 and *p* = 0.014, respectively) No significant difference in the change from baseline to the 4th week evaluation among groups (*p* = 0.540)
Michou et al. (2014) [42]	To compare the effects of a single application of one of three neurostimulation techniques (PES, paired stimulation, rTMS) on swallow safety and neurophysiological mechanisms in chronic post-stroke dysphagia.	Primary Outcome: VFSS before and after treatment Secondary outcomes: Percentage change in cortical excitability; Oral Transit Time, pharyngeal response time, Pharyngeal Transit Time, airway closure time and upper oesophageal opening time as per VFSS	Treatment group 1 (PES): significant excitability increase immediately post-Tx in the unaffected hemisphere (real vs. sham *p* = 0.043) and in the affected hemisphere 30 min post-Tx (real vs. sham *p* = 0.04). With Paired Stimulation, cortical excitability increased 30 min post-Tx in the unaffected side (*p* = 0.043) compared to sham, and immediately post-Tx in the affected hemisphere following contralateral Paired stimulation (*p* = 0.027) Treatment group 2 (paired neurostimulation): an overall increase in corticobulbar excitability in the unaffected hemisphere (*p* = 0.005) with an associated 15% reduction in aspiration (*p* = 0.005) when compared to sham. Pharyngeal response time was significantly shorter post-treatment with real stimulation compared to sham (*p* = 0.007) Treatment group 3 (rTMS): an increase in excitability in the unaffected hemisphere, but no significant difference compared to sham. No change in the affected hemisphere. Corticobulbar excitability of pharyngeal motor cortex was beneficially modulated by PES, Paired Stimulation and to a lesser extent by rTMS.
Zhang et al. (2019) [45]	To determine whether rTMS NMES effectively ameliorates dysphagia and how rTMS protocols (bilateral vs. unilateral) combined with NMES can be optimized.	Primary outcome: Cortical excitability(amplitude of the motor evoked potential) Secondary outcomes: SSA [53] and DD [55].	Compared with group 2 or 3 in the affected hemisphere, group 4 displayed a significantly greater percentage change (*p*.0.017 and *p*.0.024, respectively). All groups displayed significant improvements in SSA and DD scores after treatment and at 1-month follow-up. The percentage change in cortical excitability increased over time in either the affected or unaffected hemisphere in treatment groups 1, 2 and 4 (*p* < 0.05). In Group 3, the percentage change in cortical excitability in the unaffected hemisphere significantly decreased after the stimulation course (*p* < 0.05). Change in SSA and DD scores in group 4 was markedly higher than that in the other three groups at the end of stimulation (*p*.0.02, *p*.0.03, and *p*.0.005) and still higher than that in group 1 at the 1-month follow-up (*p*.0.01).

Note. NMES is at motor stimulation level unless explicitly mentioned. Notes. ASHA-NOMS—American speech-language-hearing association national outcome measurement system; BI—Barthel index; CDS—clinical dysphagia scale; CT—computed tomography; DD—degree of dysphagia; DOSS—dysphagia outcome and severity scale; DSRS—dysphagia severity rating scale; DT—dysphagia therapy; EES— electrokinesiographic/electromyographic study of swallowing; EQ-5D—European Quality of Life Five Dimension; FDS—functional dysphagia scale; FEDSS—fiberoptic endoscopic dysphagia severity scale; FEES—fiberoptic endoscopic evaluation of swallowing; FOIS—functional oral intake scale; HNCI—head neck cancer inventory; IADL—instrumental activities of daily living; ICU—intensive care unit; LCD—laryngeal closure duration; LPM—laryngeal-pharyngeal mechanogram; MASA—Mann assessment of swallowing ability; MBS—modified barium swallow; MBSImp—modified barium swallow impairment profile; MDADI—M.D. Anderson dysphagia inventory; MEG—magnetoencephalography; MEP—motor evoked potentials; MMSE—mini-mental state exam; MRI—magnetic resonance imaging; mRS—modified rankin scale; MS—multiple sclerosis; NEDS—neurological examination dysphagia score; NIHSS—National Institutes of Health Stroke Scale; NMES—neuromuscular electrical stimulation; NS—not significant; OD—oropharyngeal dysphagia; OPSE—oropharyngeal swallow efficiency; OST—oral sensorimotor treatment; PAS—penetration—aspiration scale; PES—pharyngeal electrical stimulation; PESO— pharyngoesophageal segment opening; RMT— resting motor thresholdS; rTMS—repetitive transcranial magnetic stimulation; SAPP—swallowing activity and participation profile; SDQ—swallowing disturbance questionnaire; SFS—swallow function score; SHEMG— electromyographic activity of the submental/suprahyoid muscles complex; SLT—speech and language therapist; SSA—standardised swallowing assessment; SWAL-QOL—swallowing quality of life; TBI—traumatic brain injury; tDCS—transcranial direct current stimulation UPDRS—unified Parkinson’s disease rating scale; VFSS—videofluoroscopic swallowing study; WST—water swallow test.

### 3.3. Risk of Bias Assessment and Methodological Quality

The Begg and Mazumdar rank correlation procedure produced a tau of −0.036 (two-tailed *p* = 0.902) and 0.178 (two-tailed *p* = 0.536) for rTMS and tDCS, respectively. The rTMS meta-analysis incorporates data from 8 studies, which yield a *z*-value of 2.348 (two-tailed *p*-value = 0.019). The fail-safe N is 4. This means that 4 ‘null’ studies need to be located and included for the combined two-tailed *p*-value to exceed 0.050. That means there would be need to be 0.5 missing studies for every observed study for the effect to be nullified. The tDCS meta-analysis incorporates data from 8 studies yielding a *z*-value of 4.857 (two-tailed *p*-value < 0.001). The fail-safe N is 42 indicating 42 ‘null’ studies need to be located and included for the combined two-tailed *p*-value to exceed 0.050; there would be need to be 5.3 missing studies for every observed study for the effect to be nullified. Both of these procedures (i.e., Begg and Mazumdar rank correlation and fail-safe N test) indicate the absence of publication bias.

Figure 2 and Figure 3 present, respectively, the risk of bias summary per domain for all included studies combined and for individual studies, assessed using the Revised Cochrane Collaboration tool for assessing risk of bias (RoB 2) [21]. The majority of studies had low risk of bias with very few exceptions.

### 3.4. Meta-Analysis: Effects of interventions

#### 3.4.1. rTMS Meta-Analysis

Eight studies using rTMS [32,33,35,42,43,44,50,51] were included in the meta-analysis. Of these, three studies provided data for two different interventions groups [32,35,36]. Six studies were excluded as OD was not confirmed by instrumental assessment and one study was excluded as rTMS was combined with NMES.

Overall within-group analysis. Pre-post intervention effect sizes ranged from 0.085 to 2.068 (Figure 4) with seven studies showing large effect sizes (Hedges’ *g* > 0.8). Pre-post interventions produced a significant, large effect size (Hedges’ *g* = 1.038).

Overall between-group analysis. A significant, small post-intervention between-group total effect size was calculated in favour of rTMS (random-effects model: *z(7)* = 2.338, *p* = 0.019, Hedges’ *g* = 0.355, and 95% CI = 0.057–0.652; Figure 5). Between-study heterogeneity was non-significant (*Q*(7) = 6.763, *p* = 0.454).

Between-subgroup analyses. Subgroup analyses were conducted to compare time between pre- and post-intervention measurement, stimulation sites (bilateral, contra-lesional and ipsi-lesional sites), pulse ranges (low: ≤600; medium; >600 and <10,000; high: ≥10,000 pulses), stimulation frequencies (1, 5 and 10 Hz), and optional behavioural training (rTMS versus rTMS + DT; Table 4). No subgroup comparisons for outcome measures were conducted as all but one study used PAS. Studies including a longer time span between pre- and post-interventions (indicating longer stimulation times) showed increased positive effect sizes compared to one-day interventions, which showed negligible effect sizes. When comparing stimulation sites, non-significant, positive effect sizes were obtained for all three stimulation groups with large ranges in effect sizes within groups. Pulse range comparisons indicated an increased significant, positive effect for higher pulse ranges. Effect sizes were only significant for large numbers of pulses delivered. Sub-analyses comparing stimulation frequencies did not indicate obvious tendencies between groups. rTMS in combination with DT showed non-significant, small positive effect sizes in one study, whereas DT alone showed similar significant, small effects sizes.

#### 3.4.2. tDCS Meta-Analysis

A total of eight studies using tDCS in stroke patients were included in the meta-analysis [27,29,36,37,38,39,40,41]. One study was excluded as having too few data for meta-analysis [31].

Overall within-group analysis. The overall pre-post intervention effect size was 1.385, with effect sizes ranging from 0.432 (small effect) to 3.365 (high effect; Figure 6). Studies showed small (*n* = 2), moderate (*n* = 1), and high effect sizes (*n* = 5).

Overall between-group analysis. A moderate but significant post-intervention between-group total effect size in favour of tDCS was found using a random-effects model (*z*(7) = 3.332, *p* = 0.001, Hedges’ *g* = 0.655, and 95% CI = 0.270–1.040; Figure 7). Between-study heterogeneity was significant (*Q*(7) = 15.034, and *p* = 0.036), with *I^2^* showing that heterogeneity accounted for 53.4% of variation in effect sizes across studies.

Between subgroup analyses. Subgroup analyses were conducted comparing time between pre- and post-intervention measurements, outcome measures, total stimulation times and stimulation current (Table 4). Increasing the number of days between pre- and post-intervention showed a strong tendency towards increased positive effect sizes, with significant effect sizes for two and four-week periods. Comparisons between measures resulted in significant, large positive effect sizes for visuoperceptual evaluation of instrumental assessment, but negligible effects when using an oral intake measure. Effect sizes for comparisons between total stimulation times indicated increased effects when using longer stimulation times. Significant, large effects were demonstrated for stimulation times of 300 min and longer. Additionally, higher stimulation currents resulted in increased significant, large positive effect sizes.

## 4. Discussion

This systematic review (Part II) aimed to determine the effects of rTMS and tDCS in people with OD. This systematic review and meta-analysis of RCT studies were completed in accordance with PRISMA procedures [19,20]. No populations were excluded based on medical diagnoses.

### 4.1. Systematic Review Findings

Like the systematic review on effects of NMES and PES in people with OD (Part I) [75], methodological problems were identified relating to unclear definitions of OD and differences in methods of confirming the presence of OD (i.e., using instrumental assessment, patient self-report or clinical assessment). Consequently, to reduce heterogeneity in participant characteristics between RCTs, only studies using instrumental assessment to confirm diagnosis of OD were included in meta-analyses. As most studies included stroke patients only, no meta-analysis could be performed to determine effects per medical diagnosis.

With the exception of one study [33], all rTMS studies included in the meta-analysis used the PAS to evaluate intervention effects. For the tDCS studies, as heterogeneity in outcome measures was larger, data on three different clinical outcome measures were used when conducting the meta-analysis. All rTMS studies used sham stimulation as a comparison group with the exception of one study which included a rTMS plus DT group [33]. For the tDCS studies, all but one study [31] combined neurostimulation with simultaneous DT. When comparing the degree of heterogeneity in study designs between brain neurostimulation (i.e., rTMS and tDCS) and peripheral neurostimulation (i.e., NMES and PES), those in the peripheral neurostimulation group were more diverse, creating greater challenges for conducting meta-analyses. Non-invasive brain stimulation studies tended to recruit smaller sample populations compared to peripheral studies [75].

#### 4.1.1. rTMS

This review prioritised reducing heterogeneity for purposes of meta-analysis. In contrast to previously published reviews that did not confirm OD by instrumental assessment, those studies were excluded from this meta-analysis. With the exception of Bath, Lee [13], earlier reviews identified significant beneficial effects of rTMS. Therefore, even though comparing the current meta-analysis with analyses from previous reviews may be challenging due to the inclusion of different outcome data, the findings from these studies seem in line with each other and this review.

#### 4.1.2. tDCS

Fewer RCTs were identified for tDCS compared with rTMS. Eight out of nine studies were eligible for meta-analysis, with one study excluded due to insufficient data; this was the only study to include non-stroke patients (presbydysphagia) [31]. Again, as previous reviews on tDCS [10,12,13,16,17,18] applied different criteria for inclusion and study methodology (e.g., differences in selection of electronic databases and publication years), final numbers of studies used for these meta-analyses ranged between two and seven publications, with reviews published before 2020 including four or fewer studies. When comparing the present results with the two most recent reviews [10,18] (both including seven studies), the beneficial effects of tDCS identified by this review were confirmed by significant, small-to-moderate effects in favour of tDCS.

#### 4.1.3. Moderators

Several factors may have had an impact on conducting meta-analyses and results. Comparing previous reviews, different decisions were made concerning criteria for meta-analyses. For example, Bath, Lee [13] excluded comparison groups with active treatment components and Chiang, Lin [12] excluded chronic stroke patients. Chronicity of stroke has shown to influence effect sizes [10,18], but selecting different primary outcomes may also result in deviating findings. For instance, Bath, Lee [13] did not find any positive effects for either rTMS or tDCS on primary outcome measures defined as death or dependency at the end of trials. Additionally, underlying medical diagnoses of OD are expected to affect meta-analyses. However, no conclusions could be drawn as very few studies of non-stroke patients were included in this review, thus no meta-analysis differentiating between diagnoses was conducted.

Similar reasons for hindering comparisons between RCTs are present in the current review, for example, spontaneous recovery and stroke severity, as were identified in the systematic review on effects of NMES and PES in people with OD (Part I) [75]. To account for the possibility of spontaneous recovery in participants, only between-subgroup meta-analyses were conducted using post-intervention data. However, the effects of stroke severity linked to OD severity remains unclear as RCTs usually did not report on the severity of stroke in sufficient detail.

Lastly, brain neurostimulation between RCTs may differ with respect to stimulation protocols (e.g., stimulation site, number and duration of treatment sessions and period) and technical parameters (e.g., frequency or number of pulses). The relatively low numbers of RCTs included in this review meant that meta-analysis could not incorporate all potential moderators. However, many of the included studies lacked sufficient details on technical parameters to allow further comparisons.

### 4.2. Limitations

Although this review followed PRISMA guidelines and aimed at reducing bias, some limitations may have had an impact on the results as presented. Only RCTs published in English were eligible in this review. Thus, some RCTs may have been excluded based on language criteria when their findings could have contributed to the current meta-analysis. Moreover, the high degree of heterogeneity between included studies hampered meta-analyses. Therefore, the results of meta-analyses and generalisations made should be interpreted with care.

## 5. Conclusions

The results of this systematic review suggest that both rTMS and tDCS show promising effects in people with OD. Meta-analysis for RCTs identified large pre-post intervention effect sizes for both types of brain neurostimulation. In addition, this analysis found significant, small and moderate post-intervention between-group effects in favour of rTMS and tCDS, respectively. However, comparisons between studies remain uncertain and challenging due to high heterogeneity in stimulation protocols and experimental parameters, potential moderators of stimulation effects, small samples sizes, and inconsistent methodological reporting.

These findings suggest that there is a need for RCTs including larger sample sizes to support future meta-analyses that will be able to adequately account for the presence of moderators. In addition, international consensus on standardised study protocols and reporting guidelines is required to support comparisons between studies.

## Figures and Tables

**Figure 1 jcm-11-00993-f001:**
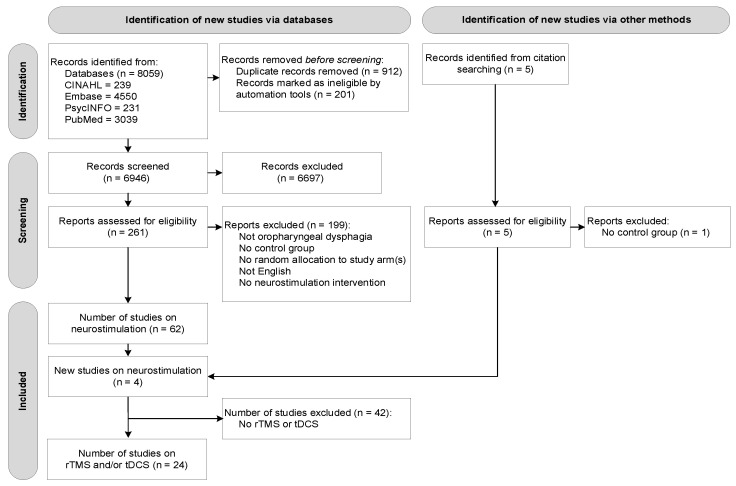
Flow diagram of the reviewing process according to PRISMA.

**Figure 2 jcm-11-00993-f002:**
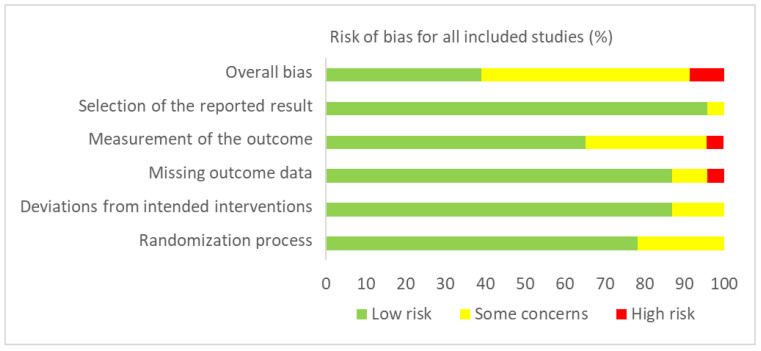
Risk of bias summary for all included studies (*n* = 24) in accordance with RoB 2 [21].

**Figure 3 jcm-11-00993-f003:**
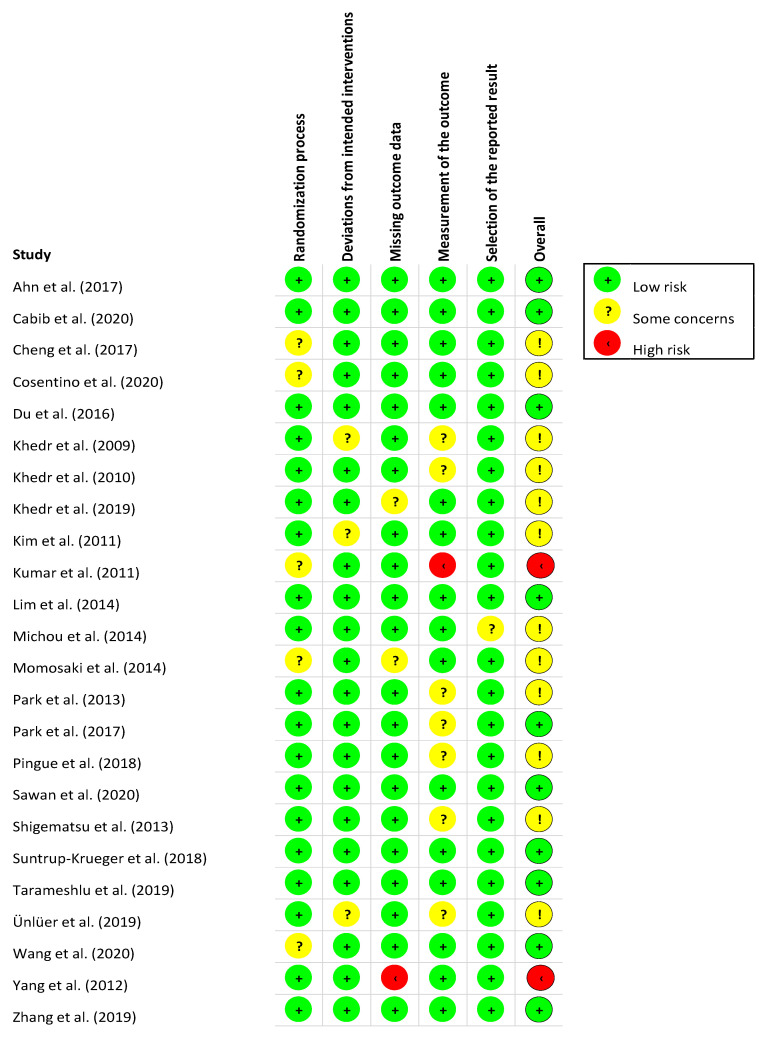
Risk of bias summary for individual studies (*n* = 24) in accordance with RoB 2 [21,27,28,29,30,31,32,33,34,36,37,38,39,40,41,42,43,44,45,46,47,48,49,50]. ***Note*.** If one or more yellow circles (domains) have been identified for a particular study, the Overall score (last column) shows an exclamation mark, indicating that the study shows some concerns (yellow circle with exclamation mark).

**Figure 4 jcm-11-00993-f004:**
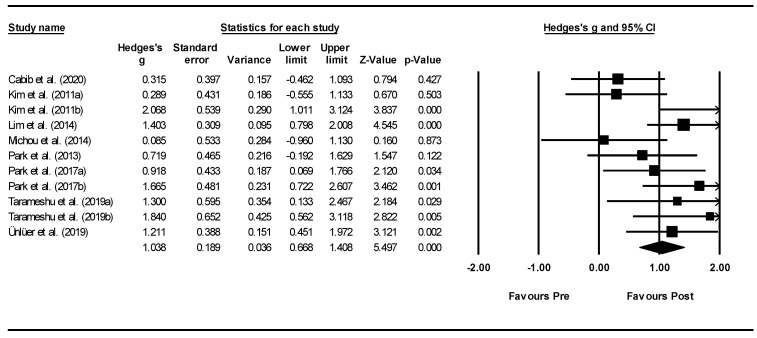
rTMS within intervention group pre-post meta-analysis [32,33,35,42,43,44,50,51]. *Notes*. Kim et al. (2011a): high frequency, Kim et al. (2011b): low frequency; Park et al. (2017a): unilateral stimulation, Park et al. (2017b): bilateral stimulation; Tarameshu et al. (2019a): rTMS, Tarameshu et al. (2019b): rTMS plus DT.

**Figure 5 jcm-11-00993-f005:**
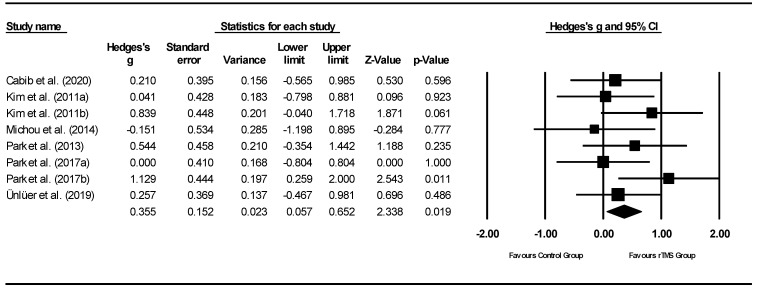
rTMS between group post meta-analysis [32,34,35,37,47,49]. *Notes.* Kim et al. (2011a): high frequency versus sham, Kim et al. (2011b): low frequency versus sham; Park et al. (2017a): unilateral stimulation versus sham, Park et al. (2017b): bilateral stimulation versus sham.

**Figure 6 jcm-11-00993-f006:**
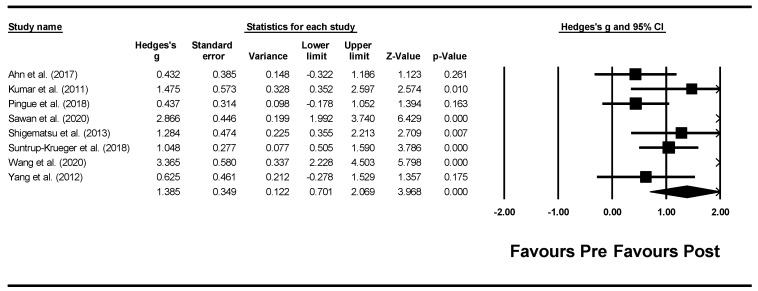
tDCS within intervention group pre-post meta-analysis [27,29,36,37,38,39,40,41].

**Figure 7 jcm-11-00993-f007:**
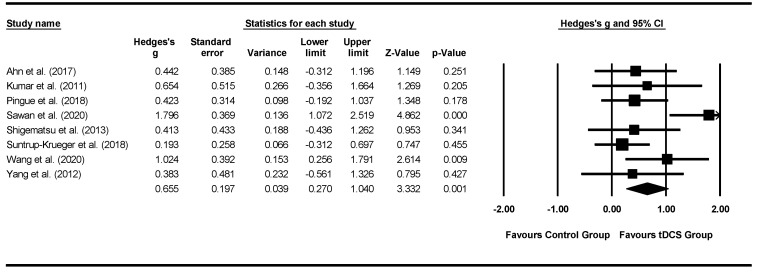
tDCS between group post meta-analysis [27,29,36,37,38,39,40,41].

**Table 1 jcm-11-00993-t001:** Search strategies.

Database and Search Terms	Number of Records
**Cinahl:** ((MH “Deglutition”) OR (MH “Deglutition Disorders”)) AND (MH “Randomized Controlled Trials”)	239
**Embase:** (swallowing/OR dysphagia/) AND (randomization/or randomized controlled trial/OR “randomized controlled trial (topic)”/OR controlled clinical trial/)	4550
**PsycINFO:** (swallowing/OR dysphagia/) AND (RCT OR (Randomised AND Controlled AND Trial) OR (Randomized AND Clinical AND Trial) OR (Randomised AND Clinical AND Trial) OR (Controlled AND Clinical AND Trial)).af.	231
**PubMed:** (“Deglutition” [Mesh] OR “Deglutition Disorders” [Mesh]) AND (“Randomized Controlled Trial” [Publication Type] OR “Randomized Controlled Trials as Topic” [Mesh] OR “Controlled Clinical Trial” [Publication Type] OR “Pragmatic Clinical Trials as Topic” [Mesh])	3039

**Table 4 jcm-11-00993-t004:** Between subgroup meta-analyses per type of neurostimulation comparing intervention groups of included studies.

Neurostimulation	Subgroup	Hedges’ *g*	*Lower Limit CI*	*Upper Limit CI*	*Z*-Value	*p*-Value
rTMS	*Time between pre-post (days)*					
	1 (*n* = 2)	0.082	−0.541	0.704	0.257	0.797
	5 (*n* = 1)	0.257	−0.467	−0.981	0.696	0.486
	14 (*n* = 5)	0.491	0.054	0.929	2.202	0.028 *
	*Stimulation site*					
	Bilateral (*n* = 2)	0.523	−0.730	1.776	0.818	0.413
	Contra-lesional (*n* = 3)	0.315	−0.141	0.771	1.353	0.176
	Ipsi-lesional (*n* = 3)	0.272	−0.251	0.795	1.020	0.308
	*Pulse range*					
	Low [≤ 600] (*n* = 2)	0.082	−0.541	0.704	0.257	0.797
	Medium [> 600 and < 10000] (*n* = 3)	0.248	−0.213	0.710	1.054	0.292
	High [≥ 10000] (*n* = 3)	0.660	0.014	1.306	2.004	0.045 *
	*Stimulation frequency (Hz)*					
	1 (*n* = 2)	0.492	−0.067	1.052	1.726	0.084
	5 (*n* = 4)	0.180	−0.257	0.617	0.809	0.419
	10 (*n* = 2)	0.552	−0.555	1.658	0.978	0.328
	*Behavioural training*					
	rTMS + DT (*n* = 1)	0.257	−0.467	0.981	0.696	0.486
	rTMS (*n* = 7)	0.375	0.031	0.720	2.135	0.033 *
tDCS	*Time between pre-post (days)*					
	4 (*n* = 1)	0.193	−0.312	0.697	0.747	0.455
	5 (*n* = 1)	0.654	−0.356	1.664	1.269	0.205
	10 (*n* = 1)	0.432	−0.192	1.037	1.348	0.178
	14 (*n* = 4)	0.784	0.056	1.512	2.112	0.035 *
	28 (*n* = 1)	1.024	0.256	1.791	2.614	0.009 *
	*Outcome measures*					
	DOSS (*n* = 5)	0.753	0.195	1.311	2.644	0.008 *
	DSRS (*n* = 1)	0.193	−0.312	0.697	0.747	0.455
	FDS (*n* = 2)	0.764	0.147	1.381	2.428	0.015 *
	*Total stimulation time (min)*					
	80 (*n* = 1)	0.193	−0.312	0.697	0.747	0.455
	150 (*n* = 1)	0.654	−0.356	1.664	1.269	0.205
	200 (*n* = 4)	0.419	0.039	0.799	2.161	0.031 *
	300 (*n* = 1)	1.796	1.072	2.519	4.862	<0.001 *
	400 (*n* = 1)	1.024	0.256	1.791	2.614	0.009 *
	*Stimulation current (mA)*					
	1 (*n* = 6)	0.430	0.148	0.712	2.985	0.003 *
	2 (*n* = 2)	1.281	0.168	2.395	2.256	0.024 *

*Note.* * Significant. *Notes.* CI—confidence interval; DOSS—dysphagia outcome and severity scale; DSRS—dysphagia severity rating scale; DT—dysphagia therapy; FDS—functional dysphagia scale; rTMS—repetitive transcranial magnetic stimulation.

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
