# Peer review of "Neurostimulation in People with Oropharyngeal Dysphagia: A Systematic Review and Meta-Analysis of Randomised Controlled Trials—Part II: Brain Neurostimulation"

_jcm, 2022, doi:10.3390/jcm11040993_

Round 1
Reviewer 1 Report
These authors performed a systematic review and meta-analysis of randomized controlled trials using neurostimulation in people with oropharyngeal dysphagia. This is part II examining brain stimulation, specifically, rTMS and tDCS. The study is novel. The writing is clear, organized, and concise. Methodology is appropriate and follows Prisma guidelines. The results are clearly presented in the tables and figures and condensed/summarized in the text. Discussion is thoughtful and appropriate, including limitations.
Reviewer 2 Report
-The introduction is missing a short description and references on stimulation protocol and neurophysiological basis ( low frequency vs. high frequency).
-Table 2- Could the authors include the exact rTMS stimulation protocol, low or high frequency, to be consistently written in the table. Only for reference 27 is written the exact rTMS protocol (3Hz), but for other studies, only “rTMS” is written. Please write the acronym rTMS consistently through Table 2 (since full terms and acronyms are given for ref 26 and ref 27, and for other studies, the term rTMS is written. Or if in Table 2 it is not specified the low vs. high-frequency stimulation, then these should be presented consistently only in Table 3.
It is suggested to authors to include in Table 2 the cortical site of rTMS stimulation and whether it is ipsilateral or contralateral to lesion site stimulation, and also the information if the study if the studies are using TMS apparatus without navigation or with navigation; How the location is mapped on M1 according to 10-20 EEG or according to individual MRI and later incorporated into TMS device to locate the stimulation area? The authors can also include the TMS and tDCS apparatus names if given in the study, which can be helpful for interested experts if conducting clinical trials or further research. Table 3 has included stimulation protocol, but it would be better if the stimulation protocol would be in Table 2. The authors are suggested to present landscape table views for easier reading and, if possible, or to join Table 2 and Table 3 since it is a bit hard to find information of interest for single study information written separately. If using the landscape view, this could be possible. An maybe to shorten some descriptions and make them more consistent and understandable.
-Please comment in Table 2/Table 3 or somewhere in the method section the years of conduction of selected studies just to have some general understanding and overview at which years these studies were conducted and maybe to organize them in the table according to the year at which the study was conducted.
-Please give the legend acronyms in tables in alphabetic order for more effortless reading and, where possible, to include the citation/reference for specific questionnaires/scales (DSRS; MASA, Hoehn and Yarhr , etc).
- Reference 32 has the term “FMS”, is it also rTMS, and maybe to give an additional explanation in the legend. “NS” in Table 2 stands to “non-significant”? NR stands for?
-Term “Cranial Stimulation” in Table 2 is maybe better to replace with “Cortical” since it may implicate “Cranial Nerves”
-The references should be listed according to the journal guidelines.
-PAS acronym is identical for “paired-associate stimulation” and “penetration–aspiration score”. Please also check the consistent writing of acronyms across tables and text. See raw 225, 245, 279, 297, 345, PAS in table 3..etc. The authors can move one acronym and leave the other or find some solution not to have identical acronyms.
-If possible, reduce at some points the acronyms because there are too many, and it becomes hard to read.
-Figure 3/4/5/6/7 would need citation/reference included for mentioned studies, and therefore, the style would be suggested to be consistent in Tables and Figures, as well as the order of presented studies (Figure vs. Table need to have the same study order). The authors are suggested to order the studies according to the year of conducting the study (as suggested above for Table 2/Table 3). For example, in Table 2/ 3 the first and second authors are given [ref], but in Figures first author et al. (year). This should be consistently presented.
-For Figure 3 give in the legend the explanation for used symbols “!”, “<”, “?”, “+”..
-Table 4 include in table legends the full terms for acronym DOSS, DSRS, FDS, DT, .. in alphabetic order.
-row 386- 387m include the citation for supplementary files.:” This systematic review and meta-analysis of RCT studies were completed in accordance with PRISMA procedures.”
-If the acronym is used for the first time in the text, it should have the full term, please check entire manuscript, raw 68 ” reported on the effects of pharyngeal and neuromuscular electrical stimulation (PES and NMES)”, It should be ” reported on the effects of pharyngeal electrical stimulation (PES) and neuromuscular electrical stimulation (NMES)”
Round 2
Reviewer 2 Report
-Please check the references for Figure 4 (list of 11 studies), but the reference for 8 studies are written?
-Please check the references for Figure 5 (list of 8 studies), but the reference for 6 studies are written?
